# Human thalamic low-frequency oscillations correlate with expected value and outcomes during reinforcement learning

Antoine Collomb-Clerc [1], Maëlle C. M. Gueguen [1,2], Lorella Minotti[1,3], Philippe Kahane[1,3], Vincent Navarro[4], Fabrice Bartolomei[5,6], Romain Carron [6,7], Jean Regis [8], Stephan Chabardès[1,9], Stefano Palminteri [10,11] & Julien Bastin [1,11] ✉

Reinforcement-based adaptive decision-making is believed to recruit fronto-striatal circuits. A critical node of the fronto-striatal circuit is the thalamus. However, direct evidence of its involvement in human reinforcement learning is lacking. We address this gap by analyzing intra-thalamic electrophysiological recordings from eight participants while they performed a reinforcement learning task. We found that in both the anterior thalamus (ATN) and dorsomedial thalamus (DMTN), low frequency oscillations (LFO, 4-12 Hz) correlated positively with expected value estimated from computational modeling during reward-based learning (after outcome delivery) or punishment-based learning (during the choice process). Furthermore, LFO recorded from ATN/DMTN were also negatively correlated with outcomes so that both components of reward prediction errors were signaled in the human thalamus. The observed differences in the prediction signals between rewarding and punishing conditions shed light on the neural mechanisms underlying action inhibition in punishment avoidance learning. Our results provide insight into the role of thalamus in reinforcement-based decision-making in humans.

As the philosopher John Locke put it "reward and punishment are the only motives to a rational creature: these are the spur and the reins whereby all mankind is set on work and guided"[1]. Research in reinforcement learning aims at characterizing the processes through which people learn, by trial and error, to select actions that respectively maximize or minimize the occurrence of rewards or punishments[2]. Converging evidence suggests that

reward-based reinforcement learning engages a fronto-striatal circuit and the dopaminergic system[3–5]. The striatum receives inputs from both cortical and thalamic regions and is densely innervated by midbrain dopamine neurons. Information is then relayed back to the cortex through the basal ganglia, which project through the thalamus. However, there is no evidence in humans regarding how neural activity in the thalamus—a key node

[1]Univ. Grenoble Alpes, Inserm, U1216, CHU Grenoble Alpes, Grenoble Institut Neurosciences, 38000 Grenoble, France. [2]Department of Psychiatry, Brain Health Institute and University Behavioral Health Care, Rutgers University–New Brunswick, Piscataway, NJ, USA. [3]Neurology Department, University Hospital of Grenoble, Grenoble, France. [4]Sorbonne Université, Paris Brain Institute – Institut du Cerveau, ICM, INSERM, CNRS, AP-HP, Pitié-Salpêtrière Hospital, Paris, France. [5]Timone University Hospital, Sleep Unit, Epileptology and Cerebral Rhythmology, University Hospital of Marseille, Marseille, France. [6]Aix Marseille University, Inserm, Institut de Neurosciences des Systèmes, Marseille, France. [7]Timone University Hospital, Department of functional and stereotactic neurosurgery, University Hospital of Marseille, Marseille, France. [8]Neurosurgery Department, University Hospital of Marseille, Marseille, France. [9]Neurosurgery Department, University Hospital of Grenoble, Grenoble, France. [10]Laboratoire de Neurosciences Cognitives Computationnelles, Département d'Etudes Cognitives, ENS, PSL, INSERM, Paris, France. [11]These authors contributed equally: Stefano Palminteri, Julien Bastin. ✉e-mail: julien.bastin@univ-grenoble-alpes.fr

in this circuit—is associated with subject's expectations or with the experienced outcomes during learning.

Punishment avoidance learning is of equal ecological importance for organism survival and has been shown in many experimental settings to be at least as effective as reward seeking[6,7]. Critically, while the performance based on rewards or punishments exhibits comparable learning accuracies, subjects are constantly slower in punishment avoidance learning tasks[8]. This increase in reaction time is thought to reflect a manifestation of a Pavlovian bias according to which motor responses are inhibited by punishment expectations, irrespective of the appropriateness of the instrumental response[9–11].

Intriguingly, both fMRI and intracranial signals indicate that the behavioral asymmetry observed between reward-seeking and punishment avoidance is mirrored by a corresponding neural asymmetry. Specifically, BOLD signals (fMRI) or cortical broadband gamma activity (iEEG) in the ventromedial prefrontal cortex exhibit significant correlation with reward prediction errors during reward-based learning. Conversely, BOLD and broadband gamma are preferentially associated with punishment prediction errors in the anterior insula[12–15].

Despite early lesion studies in rabbits[16] suggesting the involvement of the mediodorsal and the anterior parts of the thalamus during punishment-avoidance learning, most of the animal studies surprisingly focused on reward-based learning[17–22], leaving the role of theses thalamic regions in punishment-based learning largely unexplored. The dorsomedial (DMTN) and anterior thalamic nuclei (ATN) have a critical role in the prefronto-striatal-thalamic limbic circuit, facilitating flexible behaviors through the updating of stimulus-action-response associations[16,22–24]. ATN and DMTN are both connected to the network of brain regions involved during reinforcement learning, such as the medial prefrontal, cingular and insular cortices[25]. Functional magnetic resonance imaging studies in humans consistently showed that thalamic signals correlate with the difference between predicted and actual outcomes[14,26]. Animal electrophysiological and lesion studies suggest that the ATN and the DMTN could play dissociable functions during decision-making[21,27]. On the one hand, ATN neuronal activity increases during fear conditioning[28] and plays a causal role in aversive avoidance tasks[16] while ATN lesions do not affect the learning of response-reward associations[19]. Given its strong connections with the medial prefrontal cortex, there were comparatively more studies in the functions of DMTN during reward-based tasks and instrumental behavior[17–19,22]. Yet, the precise mechanism by which ATN and DMTN modulate neural activity in the prefronto-striato-thalamic loops during reward vs. punishment-based learning processes were never explored so that this study aimed at testing whether oscillatory activity in these structures could be associated with learning signals[29,30].

The high spatiotemporal resolution necessary to disentangle human thalamic neuronal activities during such cognitive processes is unattainable with ordinary imaging tools. To address this gap, we conducted a study leveraging rare direct intracranial neural recordings obtained from the human limbic thalamus of eight patients (Table S1) with pharmacoresistant epilepsy who were implanted for deep brain stimulation of the anterior thalamus[31]. We investigated whether neuronal oscillations were associated with reinforcement-related signals at different time points during a well-validated reward-seeking and punishment avoidance learning task[6,12,13]. To achieve this, we fitted a Q-learning model to each patient's behavioral data to estimate trial-wise values of latent variables (option values and prediction errors). More specifically we looked for correlations with the computational model variables at the decision stage (option-specific value expectations) and the outcome stage (prediction errors). By combining intra-thalamic recordings with computational modeling of learning behavior, our study investigates time-resolved choice and learning signals in the human thalamus.

## Results

### Behavioral results

Local field potentials were recorded from eight pharmaco-resistant epileptic patients (Table S1) implanted bilaterally in the thalamus with deep-brain stimulation electrodes as a surgical treatment to alleviate their seizures. Electrodes had two upper contact pairs inside the anterior thalamic nucleus, with the more ventral contact pairs localized in the dorsomedial thalamic nucleus (Fig. 1a). Intra-thalamic recordings were collected while patients were performing a previously validated instrumental learning task with the instruction to maximize the monetary gains and minimize the monetary losses (Fig. 1b)[6,12,13].

Behavioral results were consistent with what was previously observed in this task (Fig. 1). Reaction times (Fig. 1c, d) were significantly shorter in the reward ($1173 \pm 164$ ms) than in the punishment ($1726 \pm 291$ ms) condition ($t_{(7)} = -3.10$, $p = 0.017$). Accuracy was on average (Fig. 1 e, f) higher than chance in both the reward ($65 \pm 0.04$, $t_{(7)} = 4.23$, $p = 0.0039$) and punishment conditions ($0.60 \pm 0.02$, $t_{(7)} = 5.13$, $p = 0.0014$) and was not different between the two conditions ($t_{(7)} = 1.68$, $p = 0.14$). To further check how well participants understood the task, we next examined the last four trials of each cue pair in every session, i.e., when the cue-action-reward association is presumed to be most effectively learned. This analysis (Fig. 1g) confirmed that in both the reward and punishment conditions, the accuracy approaches 70% and surpassed chance levels (reward: $0.71 \pm 0.06$, $t_{(7)} = 3.78$, $p = 0.0069$; punishment: $0.68 \pm 0.04$, $t_{(7)} = 4.96$, $p = 0.0016$; two-tailed paired student $t$-test). Finally, to assess whether or not our participants were better explained by a QL model compared to random responding, we compared the goodness of fit of the QL model compared to that of random responding using the Aikake Information Criterion (AIC). This confirmed that the QL model displayed a significantly lower AIC (indicating better fit, see Fig. 1h; random responding: $642 \pm 40$; QL: $732 \pm 36$; $t_{(7)} = -3.2$, $p = 0.015$; two-tailed paired student $t$-test). The observed rate of correct choices in those pharmacoresistant epileptic patients were comparable to similar studies in the field, such as those conducted with similar tasks in other clinical cohorts, such as Parkinson (~60%)[32], Tourette (~63%)[33] and, more recently, epileptic patients (~70%)[13]. Thus, patients learned similarly from rewards and punishments but took longer to choose between cues for punishment avoidance, in line with previous behavioral data from healthy subjects[7] or epileptic patients[12]. These results confirm that, although instrumental performances are similar, the decision process differs in reward-seeking and punishment-avoidance contexts in a way that is compatible with a motor inhibition induced by punishment expectation[8–10].

We next investigated the association between thalamic neural activity and reinforcement learning variables. To do so, we fitted a Q-learning model (QL) to behavioral data to estimate trial-wise option-specific expected values and prediction errors. The QL model generates choice probabilities applying a SoftMax function to option values (Q-values), which are then updated at the time of outcome via a prediction error minimization process[2,7]. Fitting the model means adjusting its two parameters (learning rate and choice temperature) to maximize the likelihood of observed choices (see Methods).

### Electrophysiological results

The neural activity of each recording site ($n = 48$ bipolar channels, see Methods) was then regressed in the time-frequency domain against both expectation and outcome signals at different time points during the task. Upon examining the post-operative CT scans images that were co-registered to Fast Gray Matter Acquisition T1 Inversion Recovery[34] (FGATIR) 3 T MRI images, it was determined that the electrodes consisted of two upper contact pairs positioned within the anterior thalamic nucleus and of the more ventral contact pairs localized within the dorsomedial thalamic nucleus (Fig. S1). Given the absence of significant differences between thalamic

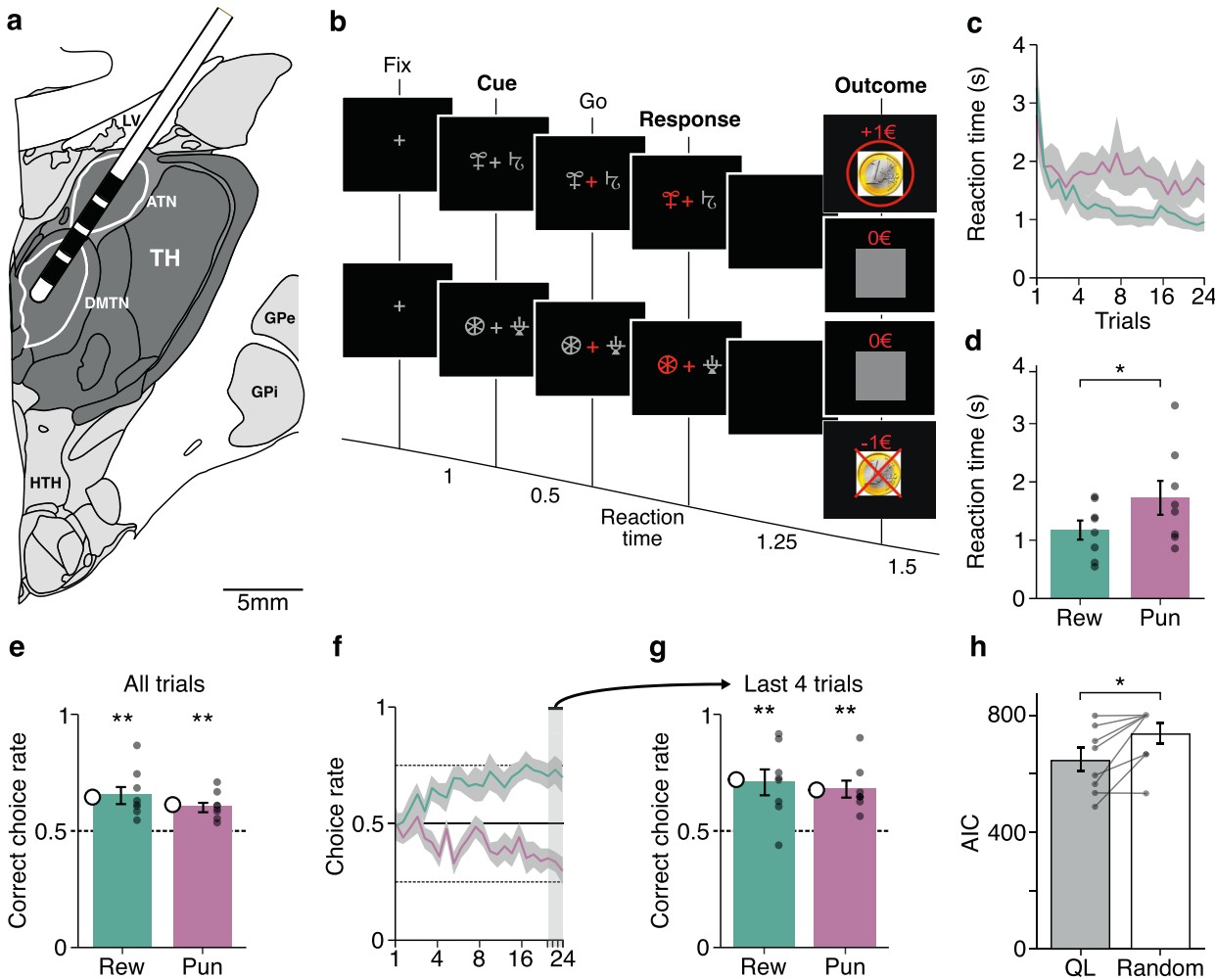

**Fig. 1 | Reinforcement-learning paradigm and behavior. a** Schematic figure (derived from Allen brain atlas) of the position of the deep brain stimulation electrodes used to record intra-thalamic signals (ATN anterior thalamic nucleus, DMTN dorsomedial thalamic nucleus, TH Thalamus, HTH Hypothalamus, GPi/GPe Globus pallidus intern/extern, LV Left ventricle). **b** Successive screenshots of a typical trial in the reward (top) and punishment (bottom) conditions. Patients had to select one abstract visual cue among the two presented on each side of a central visual fixation cross and subsequently observed the outcome. Durations are given in seconds. **c** Average±SEM reaction times across patients ($n = 8$ patients) through trials shown separately for the reward (Rew, green) and punishment (Pun, red) conditions. **d** Average ± SEM reaction times across patients ($n = 8$ patients) in the reward and punishment conditions. Dots represent data from individual patients. Asterisk indicates the significance of a paired two-sided $t$-test comparing reaction times between conditions ($t_{(7)} = -3.10$, $p = 0.017$). **e** Average±SEM choice performance across patients ($n = 8$ patients). The average predicted performance from a fitted Q-learning model is indicated by a white dot for each condition. Gray dots represent data from individual patients. Asterisk indicates the significance of the one-sample two-sided $t$-test used to compare for each condition the correct choice rate to the chance level (i.e., 50%; reward: $t_{(7)} = 4.23$, $p = 0.0039$; punishment: $t_{(7)} = 5.13$, $p = 0.0014$). **f** Average±SEM learning curves across patients ($n = 8$ patients) through trials shown separately for the reward and punishment conditions. **g** Average±SEM choice performance across patients ($n = 8$ patients) restricted to the last 4 trials of each condition. Asterisk indicates the significance of the one-sample two-sided $t$-test used to compare for each condition the correct choice rate to the chance level (i.e., 50%; reward: $t_{(7)} = 3.78$, $p = 0.0069$; punishment: $t_{(7)} = 4.96$, $p = 0.0016$). **h** Average ± SEM Akaike Information Criterion (AIC) of Q-learning (QL) model versus random choices across patients ($n = 8$ patients). Dots represent data from individual patients. Asterisk indicates the significance of the two-sided paired $t$-test used to compare the AIC of the QL model versus random choices ($t_{(7)} = -3.2$, $p = 0.015$). Source data are provided as a Source Data file.

nuclei (ATN vs. DMTN) and sides (Left vs. Right, Figs. S2, 3), in the following, all the analyses were conducted across all recording sites. This time-frequency analysis confirmed the presence of expected value signals in low frequency oscillations (LFO, 4-12 Hz) after the cue (Fig. 2a) and before the choice onset (Fig. 2b). The LFO frequency regime was preferred to separately analyzing theta (4-8 Hz) or alpha (8-12 Hz) because there was no empirical evidence for the existence of separate clusters in the time-frequency domain (Figs. 2a, b and 3a, b) and for consistency with a connected literature on the functional role of other subcortical areas during cognitive tasks (e.g. subthalamic LFO[35–39]).

We first investigated neural signals occurring after the cue (Fig. 2a) and before the choice onset (Fig. 2b). We found that low-frequency oscillations (LFO, 4–12 Hz) were significantly correlated with punishment expectations ($Q_p$) early after the cue onset (Fig. 2c; 0.85 to 1.56 s window, $\beta_{Q_p} = 0.39 \pm 0.01$, sum($t_{(47)}$) = 41.77, $p_c < 0.05$) whereas there was no significant association between thalamic LFO and reward expectation ($Q_r$) at these latencies. Furthermore, we found that LFO were associated more strongly with $Q_p$ than with $Q_r$ (Fig. 2c; 0.55 to 1.56 s window, $\beta_{Q_p}$-$\beta_{Q_r} = 0.41 \pm 0.02$, sum($t_{(47)}$) = 51.73, $p_c < 0.05$). Consistently, when neural activity was time-locked to the choice onset (Fig. 2b), there was a significant association between thalamic LFO and expectations signals during both learning conditions (Fig. 2d; −2.03 to 1.32 s window, $\beta_{Q_r} = 0.23 \pm 0.01$, sum($t_{(47)}$) = 79.29, $p_c < 0.05$; −1.25 to −0.08 s window, $\beta_{Q_p} = 0.43 \pm 0.02$, sum($t_{(47)}$) = 81.26, $p_c < 0.05$), with LFO significantly more powerfully associated with $Q_p$ than with $Q_r$

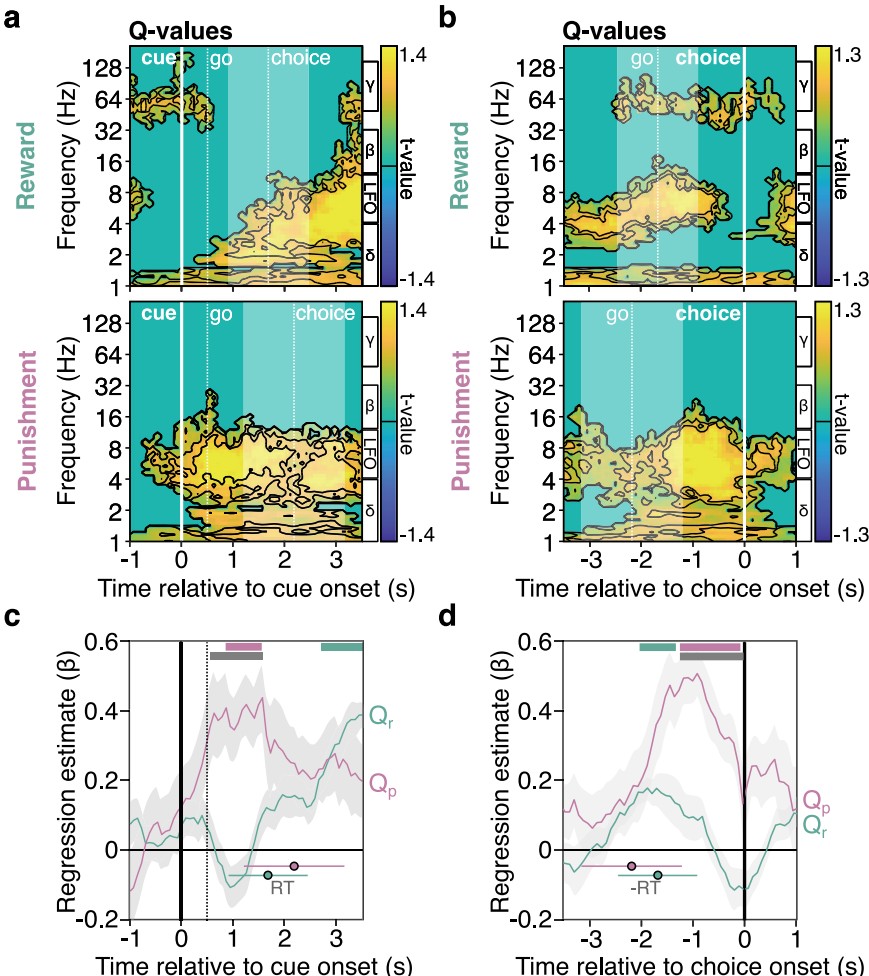

**Fig. 2 | Thalamic low-frequency oscillations associated with choice expectations during choice. a**, **b** Time-frequency decomposition of expected value signals following cue onset (**a**) or preceding participant's choice (**b**). Hotter (cooler) colors indicate more positive (negative) regression estimates (between thalamic power and Q-values). Black contours delimit statistical thresholds from pc <0.05 to pc <5.0 × 10⁻⁶. Significance was assessed using multiple two-sided one-sample student *t*-tests against zero across all thalamic sites (*n* = 48 sites). The boundaries of the frequency bands delta (δ: 1–4 Hz), low-frequency oscillations (LFO: 4–12 Hz), beta (β: 12-33 Hz), and gamma (γ: 50–150 Hz) at indicated on the right side. Shaded area represents the standard deviation of the reaction time giving the timing of the choice (**a**) or go signal onset (**b**). **c**, **d** Time-course of average (solid lines) regression estimates obtained from linear fit of LFO with $Q_r$ or $Q_p$ after the cue onset (**c**) and before the choice (**d**). Shaded gray areas around the mean represent SEM across sites (*n* = 48 sites). Colored horizontal bars displayed on the top of **c**, **d** indicate significance ($p_c$ <0.05) for one-sample *t*-tests against 0 in the reward (green) and punishment conditions (red) or for two-sided paired *t*-tests comparing the regression estimates in the reward and punishment conditions. Reaction times (RT) in the reward and punishment conditions are represented as circles (reward: green; punishment: red) and horizontal lines (mean ± sd). Source data are provided as a Source Data file.

(Fig. 2d; −1.25 to 0 s window, $\beta_{Qp}$-$\beta_{Qr}$ = 0.32 ± 0.01, sum($t_{(47)}$) = 66.52, $p_c$ < 0.05). Altogether, decision-related activities in the thalamus are consistent with a stronger encoding of punishment expectations ($Q_p$), at least during the first second after stimulus onset, although both reward and punishment expectations are encoded later on.

At the time of outcome display, we found that LFO were positively associated with expectations (Fig. 3a) and negatively associated with the magnitude of the outcome (Fig. 3b). This demonstrates that the two core components of the teaching signal—the prediction-error—are encoded by thalamic LFO which relate to the difference between what subjects expect and the actual decision outcome—what we get. Interestingly, around outcome onset, the level of expectation was significantly related to LFO only in the reward-based learning condition (Fig. 3c; 0.08 to 0.70 s window, $\beta_{Qr}$ = 0.23 ± 0.01, sum($t_{(47)}$) = 43.22, $p_c$ < 0.05). Both types of outcomes were significantly encoded by LFO in both rewarding and punishing conditions (Fig. 3d; 0.39 to 1.95 s window, $\beta_R$ = −0.21 ± 0.01, sum($t_{(47)}$) = 113.56, $p_c$ < 0.05; 0.63 to 1.80 s window, $\beta_P$ = −0.20 ± 0.01, sum($t_{(47)}$) = 83.99, $p_c$ < 0.05). These negative associations between LFO and outcomes were driven by stronger decrease of LFO when winning or losing money compared to neutral outcomes (Fig. S5).

Altogether, outcome-related activity is consistent with a similar encoding of rewards and punishments in the thalamus. Q-value encoding was detected only in the reward condition, but the absence of a significant difference between the two conditions prevent a conclusion in favor of a proper dissociation in the encoding of the prediction error. We also assess how reliable were the associations between thalamic LFO and prediction errors; we found that 7 out of 8 patients displayed a significant negative association between LFO and prediction errors (Fig. S6).

To ensure that our focus on LFO was justified, we explored activity in other frequency bands (Fig. 4). This analysis revealed that LFO were significantly associated to prediction errors in the reward ($t_{(47)}$ = −4.15, $p$ = 0.00014) and the punishment conditions ($t_{(47)}$ = −5.73, $p$ = 6.89e −07). The other frequency bands did not exhibit any significant association with prediction errors neither in the reward (delta: $t_{(47)}$ = 1.0, $p$ = 0.32; beta: $t_{(47)}$ = −0.039, $p$ = 0.97; gamma: $t_{(47)}$ = −0.61, $p$ = 0.54) nor in the punishment condition (beta: $t_{(47)}$ = 1.24, $p$ = 0.22; gamma:

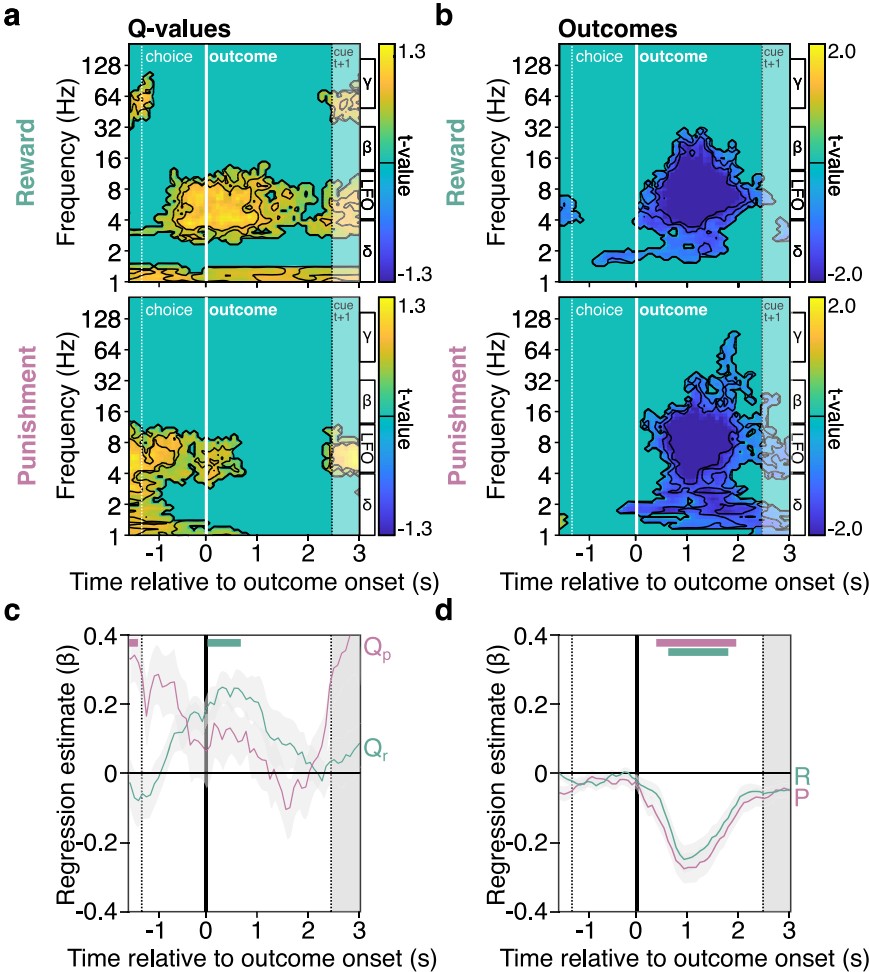

**Fig. 3 | Thalamic low-frequency oscillations associated with prediction error components. a, b** Time-frequency decomposition of prediction errors expectation (**a**: Qr or Qp) and outcome (**b**: R or P) components. Hotter (cooler) colors indicate more positive (negative) regression estimates. Black contours delimit statistical thresholds from pc <0.05 to pc <5.0 × 10⁻⁶. Significance was assessed using multiple two-sided one-sample student $t$-tests against zero across all thalamic sites ($n = 48$ sites). The boundaries of the frequency bands delta (δ: 1–4 Hz), low-frequency oscillations (LFO: 4–12 Hz), beta (β: 12–33 Hz), and gamma (γ: 50–150 Hz) at indicated on the right side. Grey shaded rectangles on the right side of all panels represent the standard deviation of the next cue pair (trial t + 1). **c, d** Time-course of average (solid lines) regression estimates obtained from linear fit of LFO with prediction error components (Qr, Qp, R, P). Shaded grₐay areas around the mean represent SEM across sites ($n = 48$ sites). Colored horizontal bars displayed on the top of c-d indicate significance ($p_c$ <0.05) for two-sided one-sample $t$-tests against 0 in the reward (green) and punishment conditions (red) or for two-sided paired $t$-tests comparing the regression estimates in the reward and punishment conditions. Source data are provided as a Source Data file.

$t_{(47)} = -0.27$, $p = 0.79$) at the exception of the delta band in the punishment condition (delta: $t_{(47)} = 2.08$, $p = 0.042$). To further check whether any frequency band could provide additional information about prediction errors, we next fitted separately reward or punishment prediction errors with all possible general linear models (GLMs) containing LFO together with any combination of other frequency bands (see Methods). Bayesian model selection designated the LFO-only GLM as providing the best account of both types of prediction errors (RPE: Ef = 0.9821, Xp = 1; PPE: Ef = 0.9821, Xp = 1). Thus, even if delta-frequency activity was significantly related to prediction errors in the punishment condition, it carried redundant information relative to that extracted from LFO.

To compare the time courses of the association between thalamic and cortical LFO and prediction errors, we re-analyzed a data-set[13] in which we recorded intracerebral data from the hippocampus, orbitofrontal and prefrontal regions during an identical task. We found that the temporal dynamics of LFO associated with prediction errors were similar between thalamic and cortical sites (Fig. S4). The main qualitative difference was that in hippocampal and cortical areas, there was an initial positive association between LFO and prediction errors which then reverted to a negative association (i.e., sign reversals) whereas this initial increase was absent in the thalamus.

## Discussion

Combining intra-thalamic human recordings with a probabilistic reinforcement learning task and trial-wise estimates of prediction errors from a Q-learning model brings a mechanistic understanding of the role of the human limbic thalamus during reward-based vs. punishment avoidance learning. We found that during the choice phase, LFO were better associated with punishment expectation signals, extending the previously observed role of the limbic thalamus in memory encoding in humans[40] to aversive contexts which were examined in rabbits in early studies[16]. These signals could originate from the dorsolateral or anterior insular cortex which were previously shown to implement punishment avoidance signals during an identical task in previous studies[12,13].

In this study, there was no significant differences between ATN and DMTN functions during learning, thus raising the question of why we were not able to detect such differences, in contrast to previous studies focusing on memory processes[40-42]. We can only offer trivial

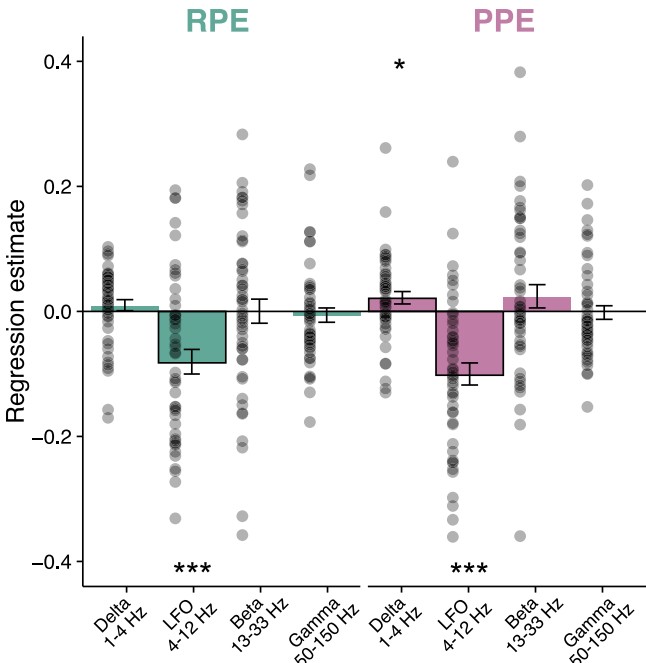

**Fig. 4 | Contribution of frequency bands to prediction error encoding in the punishment (red) and reward (green) conditions.** Average±SEM across sites of the regression estimates of power against prediction errors for the frequency bands delta (1–4 Hz), LFO (4–12 Hz), beta (13–33 Hz) and gamma (50–150 Hz). LFO power was averaged over 0–2 s post-outcome window. Stars indicate significance of regression estimates (one-sample, two-sided Student's $t$-test; LFO reward: $t_{(47)} = -4.15$, $p = 0.00014$; LFO punishment: $t_{(47)} = -5.73$, $p = 6.89e{-}07$; delta punishment: $t_{(47)} = 2.08$, $p = 0.042$). Dots correspond to regression estimates across trials for each recording site ($i = 48$ sites). RPE reward prediction error, PPE punishment prediction error. Source data are provided as a Source Data file.

explanations here, such as a relative lack of statistical power. It is also possible that we did not detect any significant difference because ATN and DMTN were equally recruited by our task. Indeed, it is very well established that during instrumental learning action-outcome associations are not the only learned variables and such learning is accompanied by concomitant habit (state-action) and Pavlovian (state-outcome) associations such that concomitant activations of the ATN and the DMTN could reflect the deployment of multiple learning systems.

Given the behavioral asymmetry in decision times between reward and punishment-based learning, we hypothesized that the neural activity could reflect the activation/inhibition balance of the thalamocortical learning circuitry during choice: the motor action threshold[40]. This interpretation is also consistent with the fact that the Pavlovian bias on reaction times has been computationally interpreted as being largely due to an increase of non-decision time, which, within the decision diffusion modeling framework, is the parameter that better captures motor inhibition[8,43].

Conversely, the association between thalamic LFO and outcomes (rewards and punishments) went in the same direction in both learning conditions (Fig. S5). The similar directionality of outcome encoding may prima facie suggest that thalamic LFO signals behavioral saliency. Yet, the (positive) correlation between thalamic LFO with the reward outcome and the (negative) correlation with the reward expectation were both observed after outcome display (Fig. 3). These opponent associations are in accordance with the very notion of reward prediction error, as it demonstrates a straightforward neural implementation of the difference between the outcome and the expectation components of the teaching signal. Furthermore, the stronger association between punishment expectation compared to reward expectation at

the time of choice (Fig. 2) also speaks against the idea that saliency alone could explain the current results.

The thalamic reward prediction error signals likely reflect a cortical input from the ventromedial prefrontal cortex / lateral orbitofrontal cortex which also exhibited sustained LFO modulations (lasting about 2.5 s), compared to the faster (<1.5 s) dynamics observed when using broadband gamma activity as a neural proxy during value rating[44] or during reinforcement learning[13]. This suggests that reinforcement learning processes trigger a sustained neural activity in the cortico-thalamic circuit involved to implement the teaching signal during reward-based learning. This also echoes recent studies in non-human primates suggesting that LFO in the orbitofrontal cortex are crucial for reward-guided learning and are driven by LFO in the hippocampus[45]. As the limbic thalamus shares extensive connections with the hippocampus, orbitofrontal, and prefrontal areas, they may form together a circuit in which reward-guided learning is encoded by LFO.

Our results also allowed us to address another open question in the field, which is to test the frequency bands involved during learning. We observed that reward prediction error was represented in the low-frequency oscillations (4–12 Hz) in the human ATN at the time of outcome onset, but this was not true for higher frequencies. In mice, beta (13–30 Hz) synchrony between the mediodorsal thalamus and the prefrontal cortex was associated with learning[17], whereas in humans, intracranial recording revealed that broadband gamma activity (50–150 Hz) recorded in the cortex encoded reward and punishment-based learning signals[13]. We speculate that this absence of association in the high gamma band in ATN/DMTN could be due to a lack of power since broadband gamma is known to be spatially more focal than LFO[46]. Our findings extend previous reports regarding the involvement of low frequency oscillations during reward-based tasks[42,47]. The (negative) correlation of thalamic LFO with the outcome and the (positive) correlation with the expectation that were simultaneously observed after outcome display in ATN/DMTN are in accordance with the very notion of a prediction error signal. These results also mirror our previous finding that in the cortex, when we used broadband gamma activity as a neural index, we found a similar opponent association between both components of prediction errors. Interestingly, the sign of the association reverted when comparing analyses based on broadband gamma and LFO in the cortex. This likely reflects the negative correlation existing between these two frequency bands, such as increased gamma power and decreased LFO accompanied local increase of the single-neuron firing rates[48,49].

Of note, evidence for punishment prediction errors encoding in the thalamus was somehow weaker, if not incomplete. If confirmed, these results could be easily accommodated by the fact that several other brain areas and systems outside the fronto-striato-thalamic circuits are devoted to punishment avoidance learning[12–15].

To conclude, our study represents a step forward in elucidating the computational reinforcement-learning processes underlain by the thalamus. Given the centrality of this brain structure within the fronto-striatal circuit, we believe that understanding its function will prove useful to computationally characterize cognitive deficits observed in many neuropsychiatric disorders[50].

## Methods
### Patients and surgical approach
Intracerebral recordings were obtained from 8 patients (38.1 ± 3.7 years old, 3 females, see demographical details in Table S1) suffering from intractable epilepsy. They were implanted bilaterally in the limbic thalamic nuclei within the anterior thalamic nuclei (ATN) with deep-brain stimulation electrodes (Medtronic DBS lead model 3389, 4 contacts, 1.5 mm wide with 0.5 mm spacing edge to edge between contacts) as a surgical treatment to alleviate their seizures.

The stereotaxic trajectory of the electrode was calculated pre-operatively based on the patient's imaging data. Electrodes were implanted through the ATN to ensure its maximal recording, with at least the two most dorsal contacts inside the ATN. As a result, the more ventral-proximal contacts pointed towards the dorsomedial thalamic nuclei (DMTN) located below the ANT along the implantation trajectory. All electrodes' positions were checked intraoperatively using a 3D X-ray image using the O-ARM tool (Medtronic, Dublin, Ireland). To improve targeting accuracy, post-operative 3D CT-scan, was obtained and merged using ROSA software with pre-operative CT-scan, T1 and nd Fast Gray Matter Acquisition T1 Inversion Recovery[34] (FGATIR) 3 T MRI images. Patient-specific segmentation of thalamic nuclei and electrode localization was done with SureTune (Medtronic, Dublin, Ireland). Briefly, this tool allowed us to fit an atlas to the patient MRI[51] while the neurosurgeon (S. Chabardes) also performed a manual segmentation to further refine anterior thalamus targeting. As a result, the more ventral-proximal contacts pointed towards the dorsomedial thalamic nuclei (DMTN) located below the ANT along the implantation trajectory. Electrode implantation was performed according to the clinical procedures of the clinical trial "France" (NCT02076698), with targeted structures preoperatively selected according strictly to clinical considerations with no reference to the current study. Patients were investigated either in the epilepsy departments of Grenoble or Marseille. Written informed consent was obtained from all participants, and the study was approved by the ethics committee (Comité de Protection des Personnes Sud-Est I, protocol number: 2011-A00083-38) in accordance with the principles of the Declaration of Helsinki.

### Behavioral task
Patients performed a probabilistic instrumental learning task. No seizures took place during the testing sessions. Patients were provided with written instructions (reformulated orally if necessary) stating that the goal was to maximize their financial payoff by considering reward-seeking and punishment avoidance as equally important. Patients performed short training sessions to familiarize themselves with the timing of events and with response buttons. Participants performed up to 6 sessions (see Table S1). Each session was an independent task containing four new pairs of cues to be learned, each pair of cues being presented 24 times for a total of 96 trials. Cues were abstract visual stimuli taken from the Agathodaimon alphabet. The four cue pairs were divided into two conditions (2 pairs of reward and 2 pairs of punishment cues), associated with different pairs of outcomes (winning 1€ versus nothing or losing 1€ versus nothing). To win money, patients had to learn by trial and error the cue-outcome associations and choose the most rewarding cue in the reward condition and the less punishing cue in the punishment condition. The reward and punishment conditions were intermingled in a learning session and the two cues of a pair were always presented together. Within each pair, the two cues were associated with the two possible outcomes with reciprocal probabilities (0.75/0.25 and 0.25/0.75). On each trial, one pair was randomly presented, and the two cues were displayed on the left and right of a central fixation cross, their relative position being counterbalanced across trials. The subject was required to choose the left or right cue by using their left or right index to press the corresponding button on a joystick (Logitech Dual Action). Since the position on the screen was counterbalanced, response (left versus right) and value (good versus bad cue) were orthogonal. The chosen cue was colored in red for 250 ms and then the outcome was displayed on the screen after 1000 ms. Visual stimuli were delivered on a 19-inch TFT monitor with a refresh rate of 60 Hz, controlled by a PC with Presentation 16.5 (Neurobehavioral Systems, Albany, CA).

### Local field potentials acquisition and processing
Intracranial signals recordings were performed at the bedside of patients from externalized electrode leads in the two days following electrode implantation (i.e., before the electrodes were connected to the stimulator). LFP signals were recorded with lead extensions connected to an EEG acquisition system (Micromed SD MRI, bandwidth 0.1–200 Hz, sampling rate 1024 or 2048 Hz). Each DBS electrode consisted of 4 contacts with a length of 1.5 mm, separated by 0.5 mm (deep brain stimulation macro-electrode 3389, Medtronic, Minneapolis, US). Signal processing was performed using a longitudinal bipolar montage between the 3 adjacent pairs of contacts per electrode to maximize the sensitivity to local sources of LFP. Overall, 48 bipolar channels were recorded (3 contact-pairs/electrode × 2 hemispheres × 8 patients) using a commercial video-EEG monitoring system (System Plus, Micromed).

Time-frequency analyses were performed with the FieldTrip toolbox (v. r7276) for MATLAB (2016). The electrophysiological data were resampled at 512 Hz and segmented into epochs from 5 s before to 5 s after the cue onset and outcome onset. A multi-tapered time-frequency transform allowed the estimation of spectral powers (Slepian tapers; lower-frequency range: 1–32 Hz, 6 cycles and 3 tapers per window; higher frequency range: 32–200 Hz, fixed time-windows of 200 ms, 4–31 tapers per window). This approach uses a steady number of cycles across frequencies up to 32 Hz (time window durations, therefore, decrease as frequency increases) whereas, for frequencies above 32 Hz, the time window duration is fixed with an increasing number of tapers to increase the precision of power estimation by increasing smoothing at higher frequencies. Time-frequency power was converted into dB (decimal logarithm transformation) to improve the Gaussian distribution of the data and thereafter baselined using a trial wise z-score transform (using the average power in a 10 s time window centered on the event of interest), as previously described[13,52]. To remove artifacts and pathological interictal epileptiform discharge, we employed the following procedure for each recording site. A sliding window of 50 ms was employed within each event of interest (10 s time window centered on each event). Trials exhibiting a power that sporadically surpassed five times the standard deviation of the average signal were excluded. Consequently, an average of 6.1% of trials per patient were excluded from each epoching window. This exclusion rate primarily stems from two patients, who had an average of 20.6% of trials excluded.

### Contributions of frequency bands
To assess the contribution of the different frequency bands to prediction error representation, reward prediction errors (RPE) or Punishment prediction errors (PPE) were regressed separately across trials against power P (normalized envelope) of each frequency band, averaged over time between 0 and 2 s after outcome onset:

$$RPE = \alpha + \beta P + \epsilon \tag{1}$$

with α corresponding to the intercept and ε to the error term. The significance of the regression estimates β was assessed across recording sites using two-sided, one-sample, Student's t-tests.

In order to determine whether other frequency bands provided additional information relative to LFO, the following 8 GLMs were compared:

$$RPE = \beta_{LFO} \times P(LFO) + \beta_{\delta} \times P(\delta) + \beta_{\beta} \times P(\beta) + \beta\gamma \times P(\gamma) \tag{2}$$

Here, $\beta_{LFO}$, $\beta_{\delta}$, $\beta_{\beta}$ and $\beta\gamma$ correspond to the regression estimates of the power P in the LFO (4–12 Hz), delta (1–4 Hz), beta (13–33 Hz), and gamma (50–150 Hz) bands in the 0–2 s time-window after the outcome onset. We compared this general linear model with eight possible

alternative models:

$$RPE = \beta_{LFO} \times P(LFO) \tag{3}$$

$$RPE = \beta_{LFO} \times P(LFO) + \beta_\delta \times P(\delta) \tag{4}$$

$$RPE = \beta_{LFO} \times P(LFO) + \beta_\beta \times P(\beta) \tag{5}$$

$$RPE = \beta_{LFO} \times P(LFO) + \beta\gamma \times P(\gamma) \tag{6}$$

$$RPE = \beta_{LFO} \times P(LFO) + \beta_\delta \times P(\delta) + \beta_\beta \times P(\beta) \tag{7}$$

$$RPE = \beta_{LFO} \times P(LFO) + \beta_\delta \times P(\delta) + \beta\gamma \times P(\gamma) \tag{8}$$

$$RPE = \beta_{LFO} \times P(LFO) + \beta_\beta \times P(\beta) + \beta\gamma \times P(\gamma) \tag{9}$$

$$RPE = \beta_{LFO} \times P(LFO) + \beta_\delta \times P(\delta) + \beta_\beta \times P(\beta) + \beta\gamma \times P(\gamma) \tag{10}$$

The model comparison was conducted using the Variational Bayesian Analysis (VBA) toolbox[53]. Log-model evidence obtained in each recording site was taken to a group-level, random-effect, Bayesian model selection (RFX-BMS) procedure[54]. The RFX-BMS provided an exceedance probability (Xp) that measures the likelihood of a given model being more frequently implemented relative to all the others considered in the model space in the population from which samples are drawn.

### Behavioral analysis and modeling
The percentage of correct choice (i.e., selection of the most rewarding or the less punishing cue) and reaction time (between cue onset and choice) were used as dependent behavioral variables. Statistical comparisons between the correct choice rate and chance choice rate (i.e., 0.5) were assessed using $t$-tests. Statistical comparisons of correct choice rate and reaction times between reward and punishment conditions were assessed using paired $t$-tests.

A standard Q-learning algorithm (QL) was used to model choice behavior. For each pair of cues, A/B, the model estimates the expected value of choosing A (Qa) or B (Qb), according to previous choices and outcomes. The initially expected values of all cues were set at 0, which corresponded to the average of all possible outcome values. After each trial (t), the expected value of the chosen stimuli (say A) was updated according to the rule:

$$Qa_{t+1} = Qa_t + \alpha * \delta t \tag{11}$$

The outcome prediction error, $\delta(t)$, is the difference between obtained and expected outcome values:

$$\delta_t = R_t + Qa_t \tag{12}$$

with R(t) the reinforcement value among −1€, 0€, and +1€. Using the expected values associated with the two possible cues, the probability (or likelihood) of each choice was estimated using the SoftMax rule:

$$Pa_t = e^{Qa_t/\beta}/(e^{Qa_t/\beta} + e^{Qb_t/\beta}) \tag{13}$$

The constant parameters $\alpha$ and $\beta$ are the learning rate and choice temperature, respectively. Expected values, outcomes, and prediction errors for each patient were then $z$-scored across trials and used as statistical regressors for electrophysiological data analysis.

### Regression between electrophysiological signals with reward and punishment learning behaviors
Power (Y) at each time-frequency point was regressed using a general linear model against both outcome value (R) and expected value (Q) to obtain a regression estimate for each time-frequency point and each contact pair:

$$Y = \alpha + \beta_R * R + \beta_Q * Q \tag{14}$$

with $\beta_R$ and $\beta_Q$ corresponding to the R and Q regression estimates, respectively.

The significance of regression estimates across thalamic sites was assessed at each time-frequency point using a one-sample two-tailed $t$-test against zero across all bipolar channels. Statistical significance was assessed through permutation tests as previously. First, the pairing between neural responses and predictors across trials was shuffled randomly 300 times for each recording site. Second, we performed 60,000 random combinations of all sites, drawn from the 300 shuffles calculated previously for each site. The maximal cluster-level statistics (the maximal sum of $t$-values over contiguous time points exceeding a significance threshold of 0.05) were extracted for each combination to compute a "null" distribution of effect size. The $p$-value of each cluster in the original (non-shuffled) data was finally obtained by computing the proportion of clusters with higher statistics in the null distribution, and reported as the corrected $p$ value noted ($p_c$).

Low-frequency (4–12 Hz) time series were computed, and the same general linear model approach was used for each time point of the time series separately in the reward and punishment conditions. The significance of regressors was assessed using a cluster correction approach comparable to the one described above.

### Reporting summary
Further information on research design is available in the Nature Portfolio Reporting Summary linked to this article.

## Data availability
The behavior and neural data generated in this study have been deposited in the Figshare database [https://doi.org/10.6084/m9.figshare.23659896]. Source data are provided with this paper.

## Code availability
The custom codes used to generate the figures and statistics are available from the lead contact (JB) upon request.

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

## Acknowledgements

S.P. is supported by the European Research Council under the European Union's Horizon 2020 research and innovation program (ERC) (RaReMem: 101043804), and the Agence National de la Recherche (CogFinAgent: ANR-21-CE23-0002-02; RELATIVE: ANR-21-CE37-0008- 01; RANGE: ANR-21-CE28- 636 0024-01). J.B. is supported by Agence National de la Recherche (DECID: ANR-17-CE37-0018; CausaL: ANR-18-CE28-0016;; EPICOG: ANR-22-CE17-0057).

## Author contributions

J.B. conceived the study. L.M., P.K., V.N., F.B., R.C., J.R., S.C. participated to patient recruitment and inclusion. J.R., S.C. implanted the patients. M.C.M.G., J.B., L.M., P.K., V.N., F.B., R.C., J.R., S.C. participated to data acquisition. A.C.C., M.C.M.G., J.B., S.P. analyzed behavioral and neural data. A.C.C., J.B. and S.P. wrote the manuscript. All authors discussed the results and implications and commented on the manuscript at all stages.

## Competing interests

The authors declare no competing interests.
