## [Peer Review File · Nature Communications]

Human thalamic low-frequency oscillations correlate with expected value and outcomes during reinforcement learningREVIEWER COMMENTS

Reviewer #1 (Remarks to the Author):

The authors investigated the role of the thalamus in human reinforcement learning based on rarely available intracranial electrophysiological recordings from the anterior and dorsomedial thalamic nuclei (ATN, DMTN). The study poses a novel and interesting question regarding the role of the thalamus in learning, and the findings suggest that low frequency oscillatory electrophysiological activity in the thalamus is associated with expectation of reward and punishment in reinforcement learning. The authors pooled the recordings from the ATN and DMTN on the basis of not detecting a significant difference between measures from them, however. The different anatomical connectivity of the ATN and DMTN suggests different roles for these nuclei, and the absence of a difference could reflect lack of adequate power or the choice of parameters. Moreover, differences have been previously reported between the engagement of left and right ATN and DMTN during cognitive processing. I would therefore suggest that the analyses be performed separately for the left and the right ATN and DMTN, rather than pooling the data from all of the recording sites. Also, the authors discuss the frequency specificity of their findings, but a direct comparison between different frequencies was not presented. It is also not clear why the authors combined the alpha and frequency band ranges for their analyses. Finally, more detail is needed regarding the analysis methods and statistical testing applied. While a separate Methods section is provided at the end of the manuscript, more information would be helpful earlier in the manuscript at several points. I have indicated these places below, as well as made further suggestions.

Abstract

It would be helpful to present briefly the analysis approach used and the results in the abstract.

Main text

Line 29: "would put it" implies that this is not a direct quote, but the quotations marks suggest otherwise. The book from which the quote comes should be cited.

Line 35-36: Please list the structures deemed to be involved in this circuit and cite evidence for involvement of the thalamus, including whether the evidence suggests the thalamus as a whole or particular nuclei or subnuclei. What is meant by "encodes variables"?

Line 45: What is meant by "the ventral striatum and ventromedial prefrontal cortex represent reward learning signals"? It would be helpful to state what specifically was measured and deemed to represent a learning signal.

Line 49: It would be adequate to say animal studies and save the space used in listing species.

Line 54: The clinical reason for thalamic electrode implantation in humans should be briefly stated here.

Line 58: "with computational modeling of the learning behavior results" This is the first mention of computational modelling. The approach and its purpose should be stated briefly here.

Line 60: The term "pharmacoresistant" would be preferable.

Figure 1: How were the electrode locations confirmed post-operatively? Precise placement is likely to differ between the eight patients, as electrode trajectories are planned on an individual basis to avoid major vessels. It would be helpful to show the post-operative electrode localization based on individual structural imaging for at least one patient to demonstrate how the authors confirmed the electrode locations post-operatively.

Line 78: A brief explanation of what is meant by a Q-learning model is needed here. The authors state that they fitted a Q-learning model, but Q-learning is a model-free reinforcement learning algorithm.

Line 79: How were there 48 recording sites? Eight patients each had bilateral implantation of electrodes with 4 contacts each, suggesting 64 recordings sites. Reference to bipolar re-referencing first appears at line 241. Presumably the authors meant 48 bipolar channels?

Line 81: What is meant by "the absence of significant differences between sites located within the anterior thalamic nucleus (n=16 sites), the dorsomedial thalamic nucleus (n=16 sites, Supplementary. Fig. S1) or sites localized in-between (n=16)"? The anterior and dorsomedial nuclei have different anatomical connectivity, consistent with engagement in different networks. What neural activity measure was used? Were the data baselined?

Figure 2: Labelled colorbars are needed. Is power shown or T-values? The axes should be extended to show the significant clusters in their entirety. A correction should be applied to take account of recording from 4 structures: left and right ATN and left and right DMTN. What is meant by "significant clusters (cluster-corrected, $p < 0.05$) in the time domain for a one-sample t-test against 0"?

Figure 3: Line 185: Please define PE.

Line 87: How was Q calculated? A brief explanation would be helpful here.

Line 132: "LFOs oscillations". Word repetition; O stands for oscillations already.

Line 139: Typo: "are" rather than "and"?

Line 140: The authors appear to have tested only the 4-12 Hz frequency band., so it is not clear how they addressed "another open question in the field, which is to test the frequency bands involved during learning". To address the question as to which frequency bands are involved, direct comparisons should be made between frequency bands.

Methods

Line 200: The authors mention pre-operative planning of the electrode locations, but how were the actual locations confirmed post-operatively?

Line 212: The authors report that no seizures took place during the testing sessions. Were the data inspected for interictal epileptiform discharges, which could have a potential influence on attention?

Line 207: "Patients were investigated either in the epilepsy departments of Grenoble, Paris, or Marseille." Paris is not mentioned in Table S1.

Line 208: Reference should be made to the ethical standards applied. Was the Declaration of Helsinki considered?

Report

I was not able to find the data in the figshare repository. Could there be a typo in the link provided?

Reviewer #2 (Remarks to the Author):

The role of the thalamus in human reinforcement learning

In this paper, Collomb-Clerc et al. report on a rare and valuable set of intracranial recordings from the thalamus of intractable epilepsy patients. Patients completed an instrumental learning task in which they had to maximize monetary gains by learning the reward contingencies associated with different abstract visual cues, in both reward (+\$) and punishment (-\$) conditions. The authors use a computational framework (Q-learning) to study the representation of reward signals in electrophysiological activity in two different thalamic subregions. Using a multi-frequency resolved regression approach, the authors propose that low-frequency activity (mostly in the 4-12Hz range) reflects expected values and outcomes at different times in the task, with different encoding profiles for reward and punishment. The authors conclude that thalamic low frequency oscillations encode expectation and reward prediction errors.

This is one of the first papers to relate intracranial signals to RL derived parameters, one of few papers to examine thalamic activity in awake behaving human patients, and the first one to do both at the same time. The choice of task is appropriate, the regression approach adequate (if incomplete – see below) and the electrophysiological methods are equally appropriate. The number of patients is about average for a study of these characteristics. There is great novelty in these thalamic recordings and the methodological combination, which has the potential to shed new light into the involvement of anterior thalamus in reward-guided behavior.

Despite these strong points, there are multiple aspects of the paper that greatly diminished enthusiasm. First, behavioral performance for most of the patients in the dataset is borderline, with many of the patients perform barely above chance in either condition (Fig. 1d). This throws into question to what extent the patients really understood the task, which in turns has serious implications for the computational models being used in the analysis. Some more strict inclusion criteria may be appropriate, or individual-level modeling that may indicate that the chosen Q-learning model does a better job of explaining the behavioral strategy than a random 50/50 model or heuristic strategies.

The second major point relates to the relationship between neural activity and computational parameters (Figs.2 and 3). There are many concerns with these results (see below), but the overall point is that the specificity of the results, and whether the activity patterns presented correspond to value computations or to more unspecific responses. There are multiple reasons for this reservation. First, the very sustained activations (primarily Fig. 2a-b) are suspect. For example, activation related to Q-values is sustained for over 3s (Fig. 2a), which is a very long time and not in line with similar value-related activations elsewhere (to use an example from the author's see Lopez-Persem et al 2020) which are noticeably shorter. It is of course possible that the timecourse of thalamic activation is very different from cortical ones, but this may be unlikely given that cortex is presumably driving this activity. Even taken at face value, this sustained activations raise many concerns, including the fact that cross-trial correlation in Q values may bleed through across trials. The activation has not died down (and in fact looks close to maximal levels) towards the end of the window in 2b – bleeding well into the following trial given the experimental design (Fig.1). How much longer does this last? Is there an effect of trial t onto $t+1$ activity or regression results? This is especially concerning given that Q_{vt} and Q_{vt+1} are likely highly correlated across consecutive trials as the model learns values over time. Secondly, it is possible that the computational timecourses are driven not by the specific computations presented but by coarser value signals. In figs. 2a and 2b, the regression incorporates both punishment and reward blocks. I'm not convinced that is appropriate given the differences in the time course of the regression estimates (Fig.1c and d), and it raises the possibility that simple win vs loss expectations (or contextual signal) could drive the results. Perhaps not a very strong possibility given that patients don't do well in the task, but I would still suggest showing separate regressions for reward and punishment conditions. An even simpler potential explanation that should be formally ruled out is that the activations shown are simply motor or visual. Rather than reflecting value-based quantities, these activations could simply be stimulus-evoked activity (e.g. visual), something that has been previously reported in human thalamus (<https://www.nature.com/articles/s41598-021-96588-x>). Even though the activations are regression results and not simple time-locked power modulations, depending on exactly how the

Q-values and outcomes regressors look this could be a seriously confounding issue. This could be potentially addressed by having dummy regressors for confounds (motor, visual) in the regression model in addition to the regressors of interest. Similarly, related to the win/loss possibility – if this was driven by win/loss encoding (which is known to engage low f in humans, e.g. Marco-Pallares et al 2008) rather than parametric Q_v or RPE encoding,, a regression model that includes wins/losses in addition to the Q values would partially address this.

There are other aspects of the data that seem to contradict the authors' interpretation that these activations reflect value. The clearest one is presented in Fig. 3d (and Fig. S2): reward and punishment regression estimates look basically identical in sign, magnitude and timecourse. If this was a value signal, wouldn't one expect neural responses to rewards and punishments to be opposite in sign? Again, these responses look very unspecific and generate significant doubts into the reward interpretation.

A relatively minor point – the regression estimates in Fig, 1c and 1d seem pretty tight (low SEM) given that these analysis include all the electrodes in the sample. I would be curious to know what the differences in encoding is across patients (esp. in light of behavioral variability) and electrodes. I would be surprised if these activations were so robust that they appeared in every thalamic electrode, but it would be good to know either way.

In a different but related line, it would be very valuable to look at the time frequency representations (raw power, without the computational analysis) to see if there are power modulations at the time where the computational associations may be happening.

Finally, the authors report no activation in relatively higher frequencies, contrary to what has been described in cortical recordings. I realize that because of sampling rate limitations it would be difficult to try to extend analysis into the high gamma band, and the authors cautiously (and I think correctly) limit their analyses to $<128\text{Hz}$, but even that should be able to capture gamma and some broadband high gamma activations. This is surprising enough that it merits some discussion - do they have an interpretation for this?

The discussion section was very short and lacking in detail on the implications of the findings. Very little space is devoted to discussing the proposed notion that low frequency oscillations encode reward computations. This is a pretty novel idea, and a departure from the more conventional role that high frequency oscillations encode local computations for which significantly more evidence exists, and so I would like to see a direct discussion of the potential neurobiological implications, relationship to high frequency activity, etc. More importantly – taken at face value, the results implicate that there are separable bases for expected value and outcome value computations, but how these are combined to give rise to RPE signals (SFig.2) is ignored (a related point – RPEs [Fig. 2a] and outcomes [Fig. 3b] look almost identical, which relates to my point above about specificity). If that is correct, one could expect some systematic relationship between the neural activity underlying Q -values (Fig 3a) and outcomes (Fig 3b) that mirrors (or not) their algorithmic relationship (equation 2), but whether this is the case is not discussed at all. This mechanistic type of insight is where this approach has the potential to really advance the field – not devoting time to this seems like a missed opportunity.

Overall, whereas there is potentially high value and novelty in this dataset, there are significant concerns about the quality of the behavioral data, the specificity of the neural responses that would require significant reanalysis of the data. Similarly, the incomplete discussion would need to be significantly reworked.

Minor comments

- Fig.1D and E – significance asterisks look like datapoints and are confusing – suggest differentiating in some other way (maybe coloring datapoints).
- How was the data baselined, if at all? Probably not critical for regression analysis but it should be specified; there is a vague mention of z-scoring but very little details. Was this done within f band, or some other way?
- There is an asymmetry in learning from reward vs punishment (Fig.1) – do the authors have any interpretation or explanation for this?

- More details are needed regarding the number of electrodes in the sample. 8 patients received DBS leads with 4 contacts each, but there are 48 datasets in the final sample. Did some patients undergo bilateral implantation and had 8 contacts instead of 4? If all patients were bilateral, were any electrodes discarded? If so, what was the quality control criterion for exclusion?
- Fig.2 a and b have no color scales, which makes it very hard to estimate the strength of the reported activations. And what do the contour lines represent? I assume increasing significance thresholds, but this is not explained.
- Author's first and last names order seems to change.
- l132 – "LFOs oscillations" is redundant

Color Code:

Black: reviewers' comments

Blue: authors' responses

Green: new sections/updates in the manuscript

Reviewer #1

The authors investigated the role of the thalamus in human reinforcement learning based on rarely available intracranial electrophysiological recordings from the anterior and dorsomedial thalamic nuclei (ATN, DMTN). The study poses a novel and interesting question regarding the role of the thalamus in learning, and the findings suggest that low frequency oscillatory electrophysiological activity in the thalamus is associated with expectation of reward and punishment in reinforcement learning.

R1.1. The authors pooled the recordings from the ATN and DMTN on the basis of not detecting a significant difference between measures from them, however. The different anatomical connectivity of the ATN and DMTN suggests different roles for these nuclei, and the absence of a difference could reflect lack of adequate power or the choice of parameters. Moreover, differences have been previously reported between the engagement of left and right ATN and DMTN during cognitive processing. I would therefore suggest that the analyses be performed separately for the left and the right ATN and DMTN, rather than pooling the data from all of the recording sites.

We agree that the different anatomical connectivity of the ATN and DMTN suggests dissociable decision-making functions. This is why we systematically tested for functional differences between ATN and DMTN at each phase of the task (i.e., at either cue or outcome onset) in **Figure S3** but we indeed failed to find any significant differences between the two structures. We also noticed that previous studies dissociated left and right thalamic nuclei, but we did not initially perform the analysis because the theoretical reasons justifying such a lateral dissociation were not obvious.

That said, we have now explored this issue by comparing the association between LFO and prediction errors in the left and right recording sites of the ATN and DMTN (**Figure R1/S2**). We performed a two-way ANOVA on each point of the regression estimates (between LFO and PE) time courses (separately for RPE and PPE) with structure (ATN, DMTN) and sides (Left, Right) as factors with interaction. We found no significant statistical effect on those time courses comparing either the structures, the sides, or the interaction between structures and sides.

Figure R1 illustrates that new analysis. We added this figure as **Supplementary Figure S2** of the manuscript. We also further introduce and discuss the differences (or absence of difference) between ATN and DMTN functions (see below).

Figure R1 (Supplementary Figure S2 in manuscript). Reward and punishment prediction error signals in the left (L) and right (R) ATN and DMTN. Time course of regression estimates obtained from linear fit of low frequency oscillations (4-12 Hz) with prediction errors modeled separately for the reward and punishment conditions. PPE: punishment prediction error; RPE: reward prediction error. Grey shaded areas represent inter-sites SEM. For both structures, no significant cluster (cluster-corrected, $p_c < 0.05$) in the time domain was found for the Anova comparing the regression estimates in the structures (ATN, DMTN) and the sides (Left, Right) with interaction.

Introduction Line 67: The dorsomedial (DMTN) and anterior thalamic nuclei (ATN) have a critical role in the prefronto-striatal-thalamic limbic circuit, facilitating flexible behaviors through the updating of stimulus-action-response associations^{23,24,22,16}. ATN and DMTN are both connected to the network of brain regions involved during reinforcement learning, such as the medial prefrontal, cingular and insular cortices²⁵. Functional magnetic resonance imaging studies in humans consistently showed that thalamic signals correlate with the difference between predicted and actual outcomes^{26,14}. Animal electrophysiological and lesion studies suggest that the ATN and the DMTN could play dissociable functions during decision-making^{27,21}. On the one hand, ATN neuronal activity increases during fear conditioning²⁸ and plays a causal role in aversive avoidance tasks¹⁶ while ATN lesions do not affect the learning of response-reward associations¹⁹. Given its strong connections with the medial prefrontal cortex, there were comparatively more studies in the functions of DMTN during reward-based tasks and instrumental behavior^{17,18,19,22}. Yet, the precise mechanism by which ATN and DMTN modulate neural activity in the prefronto-striato-thalamic loops during reward vs. punishment-based learning processes were never explored so that this study aimed at testing whether oscillatory activity in these structures could be associated with learning signals^{29,30}.

Results Line 136: The neural activity of each recording site (n=48 bipolar channels, see Methods) was then regressed in the time-frequency domain against both expectation and outcome signals at different time points during the task. Upon examining the post-operative CT scans images that were co-registered to Fast Gray Matter Acquisition T1 Inversion Recovery³² (FGATIR) 3T MRI images, it was determined that the electrodes consisted of two upper contact pairs positioned within the anterior thalamic nucleus and of the more ventral contact pairs localized within the dorsomedial thalamic nucleus (**Fig. S1**). Given the absence of significant differences between thalamic nuclei (ATN vs. DMTN) and sides (Left vs. Right, **Fig. S2-3**), in the following, all the analyses were conducted across all recording sites.

Discussion Line 219: In this study, there was no significant differences between ATN and DMTN functions during learning, thus raising the question of why we were not able to detect such differences, in contrast to previous studies focusing on memory processes³⁸⁻⁴⁰. We can only offer trivial explanations here, such as a relative lack of statistical power. It is also possible that we did not detect any significant difference because ATN and DMTN were equally recruited by our task. Indeed, it is very well established that during instrumental learning action-outcome associations are not the only learned variables and such learning is accompanied by concomitant habit (state-action) and Pavlovian (state-outcome) associations such that concomitant activations of the ATN and the DMTN could reflect the deployment of multiple learning systems.

R1.2. Also, the authors discuss the frequency specificity of their findings, but a direct comparison between different frequencies was not presented.

This is a good point. We acknowledge that although time-frequency analyses were provided, we did not formally compare the brain-behavior associations across all frequency bands. We now provide separate regressions with RPE or PPE across four delta (1-4 Hz), LFO (4-12 Hz), beta (13-33 Hz), and gamma (50-150 Hz) frequency bands. More precisely, for each frequency band and recording site, power time series were averaged over the outcome time window (0 - 2 s) and regressed against PE separately for the reward and punishment conditions. This analysis revealed that LFO were significantly associated to PE encoding in the reward ($t(47)=-4.15$, $p=0.00014$) and the punishment conditions ($t(47)=-5.73$, $p=6.89e-07$). The other frequency bands did not exhibit any significant association with PE neither in the reward (delta: $t(47)=1.0$, $p=0.32$; beta: $t(47)=-0.039$, $p=0.97$; gamma: $t(47)=-0.61$, $p=0.54$) nor in the punishment condition (beta: $t(47)=1.24$, $p=0.22$; gamma: $t(47)=-0.27$, $p=0.79$) at the exception of the delta band in the punishment condition (delta: $t(47)=2.08$, $p=0.042$).

Furthermore, to quantify the respective contribution of activities in the different frequency bands to PE signaling across thalamic sites, we next included them as separate regressors in general linear models meant to explain PE. To further check whether any frequency band could provide additional information about PE, we fitted separately RPE or PPE with all possible general linear models (GLMs) containing 4-12 Hz together with any combination of other frequency bands (see Materials and methods). Bayesian model selection designated the 4-12 Hz-only GLM as providing the best account of both types of prediction errors (RPE: $E_f = 0.9821$, $X_p = 1$; PPE: $E_f = 0.9821$, $X_p = 1$). Thus, even if delta-frequency activity was significantly related to PPE, it carried redundant information relative to that extracted from 4-12 Hz.

We added **Figure R2** as a new **Figure 4** to the manuscript to illustrate these new analyses. We also added specific sections within the result and method parts of the manuscript as follows:

Figure R2 (Figure 4 in manuscript). Contribution of frequency bands to prediction error encoding in the punishment (red) and reward (green) conditions. Average across sites of the regression estimates of power against prediction errors for the frequency bands delta (1-4 Hz), LFO (4-12 Hz), beta (13-33 Hz) and gamma (50-150 Hz). LFO power was averaged over 0-2 s post-outcome window. Stars indicate significance ($p < 0.05$) of regression estimates (one-sample, two-sided Student's t -test). Dots correspond to regression estimates across trials for each recording site ($n=48$). RPE: reward prediction error. PPE: punishment prediction error.

Results Line 185: To ensure that our focus on LFO was justified, we explored activity in other frequency bands (Fig. 4). This analysis revealed that LFO were significantly associated to prediction errors in the reward ($t_{(47)} = -4.15$, $p = 0.00014$) and the punishment conditions ($t_{(47)} = -5.73$, $p = 6.89e-07$). The other frequency bands did not exhibit any significant association with prediction errors neither in the reward (delta: $t_{(47)} = 1.0$, $p = 0.32$; beta: $t_{(47)} = -0.039$, $p = 0.97$; gamma: $t_{(47)} = -0.61$, $p = 0.54$) nor in the punishment condition (beta: $t_{(47)} = 1.24$, $p = 0.22$; gamma: $t_{(47)} = -0.27$, $p = 0.79$) at the exception of the delta band in the punishment condition (delta: $t_{(47)} = 2.08$, $p = 0.042$). To further check whether any frequency band could provide additional information about prediction errors, we next fitted separately reward or punishment prediction errors with all possible general linear models (GLMs) containing LFO together with any combination of other frequency bands (see Methods). Bayesian model selection designated the LFO-only GLM as providing the best account of both types of prediction errors (RPE: $E_f = 0.9821$, $X_p = 1$; PPE: $E_f = 0.9821$, $X_p = 1$). Thus, even if delta-frequency activity was significantly related to prediction errors in the punishment condition, it carried redundant information relative to that extracted from LFO.

Methods Line 363:

Contributions of frequency bands. To assess the contribution of the different frequency bands to prediction error representation, reward prediction errors (RPE) or Punishment prediction errors (PPE) were regressed separately across trials against power P (normalized envelope) of each frequency band, averaged over time between 0 and 2 s after outcome onset:

$$RPE = \alpha + \beta P + \epsilon$$

with α corresponding to the intercept and ϵ to the error term. The significance of the regression estimates β was assessed across recording sites using two-sided, one-sample, Student's t -tests.

In order to determine whether other frequency bands provided additional information relative to LFO, the following 8 GLMs were compared:

$$\text{RPE} = \beta_{\text{LFO}} \times P(\text{LFO}) + \beta_{\delta} \times P(\delta) + \beta_{\beta} \times P(\beta) + \beta_{\gamma} \times P(\gamma)$$

Here, β_{LFO} , β_{δ} , β_{β} and β_{γ} correspond to the regression estimates of the power P in the LFO (4-12 Hz), delta (1-4 Hz), beta (13-33 Hz), and gamma (50-150 Hz) bands in the 0-2 s time-window after the outcome onset. We compared this general linear model with eight possible alternative models:

$$\text{RPE} = \beta_{\text{LFO}} \times P(\text{LFO})$$

$$\text{RPE} = \beta_{\text{LFO}} \times P(\text{LFO}) + \beta_{\delta} \times P(\delta)$$

$$\text{RPE} = \beta_{\text{LFO}} \times P(\text{LFO}) + \beta_{\beta} \times P(\beta)$$

$$\text{RPE} = \beta_{\text{LFO}} \times P(\text{LFO}) + \beta_{\gamma} \times P(\gamma)$$

$$\text{RPE} = \beta_{\text{LFO}} \times P(\text{LFO}) + \beta_{\delta} \times P(\delta) + \beta_{\beta} \times P(\beta)$$

$$\text{RPE} = \beta_{\text{LFO}} \times P(\text{LFO}) + \beta_{\delta} \times P(\delta) + \beta_{\gamma} \times P(\gamma)$$

$$\text{RPE} = \beta_{\text{LFO}} \times P(\text{LFO}) + \beta_{\beta} \times P(\beta) + \beta_{\gamma} \times P(\gamma)$$

$$\text{RPE} = \beta_{\text{LFO}} \times P(\text{LFO}) + \beta_{\delta} \times P(\delta) + \beta_{\beta} \times P(\beta) + \beta_{\gamma} \times P(\gamma)$$

The model comparison was conducted using the Variational Bayesian Analysis (VBA) toolbox⁵¹. Log-model evidence obtained in each recording site was taken to a group-level, random-effect, Bayesian model selection (RFX-BMS) procedure⁵². The RFX-BMS provided an exceedance probability (X_p) that measures the likelihood of a given model being more frequently implemented relative to all the others considered in the model space in the population from which samples are drawn.

R1.3. It is also not clear why the authors combined the alpha and theta frequency band ranges for their analyses.

We understand the reviewer's concern. The main reason explaining our choice to combine alpha and theta bands was that we used a data-driven approach: time-frequency analyses revealed a significant cluster of neural activity covering the theta and alpha frequency bands. To further investigate whether LFO observed at the population level could reflect the existence of variable patterns at the individual recording site level, we checked the time-frequency data of individual contact-pairs did not exhibit any dissociable cluster of activity within alpha or theta bands by visually inspecting all time-frequency maps. None of the individual recording sites displayed separable clusters of activity in theta and alpha frequency bands. At a more conceptual level, there is indeed an interesting debate on this topic in the field of brain oscillations. The emerging consensus is to simply report the frequency extent of the studied brain oscillations since currently there is no consensus on the definition of theta, alpha, beta or gamma: the exact frequency extent of these oscillations is now clearly known to depend on a variety of factors such as the cognitive architecture, brain region, recording technique and species studied. We therefore used the term LFO for consistency with a connected literature on the functional role of the human subthalamic nucleus that uses this terminology to investigate subthalamic LFO (5-12 Hz in this field of research, e.g., Solomon et al., 2017; Scangos et al., 2018; Wojtecki et al., 2006; Brittain et al., 2012).

We included this rationale in the manuscript:

Results Line 146: The LFO frequency regime was preferred to separately analyzing theta (4-8 Hz) or alpha (8-12 Hz) because there was no empirical evidence for the existence of separate clusters in the time-frequency domain (**Fig 2a-b-3a-b**) and for consistency with a connected literature on the functional role of other subcortical areas during cognitive tasks (e.g. subthalamic LFO^{33,34,35,36,37}).

R1.4. Finally, more detail is needed regarding the analysis methods and statistical testing applied. While a separate Methods section is provided at the end of the manuscript, more information would be helpful earlier in the manuscript at several points. I have indicated these places below, as well as made further suggestions.

We apologize if there was a lack of clarity regarding how analyses were done. We edited the entire manuscript to make it easier to follow.

Minor points

Abstract: It would be helpful to present briefly the analysis approach used and the results in the abstract.

We expanded the abstract in that sense:

Abstract Line 22: Reinforcement-based adaptive decision-making is believed to recruit fronto-striatal circuits. A critical node of the fronto-striatal circuit is the thalamus, which sends critical excitatory output to the cortex. However, direct evidence of its involvement in human reinforcement learning is lacking. We address this gap by analyzing intra-thalamic electrophysiological recordings from eight participants while they performed a reinforcement learning task. We found that in both the anterior thalamus (ATN) and dorsomedial thalamus (DMTN), low frequency oscillations (LFO, 4-12 Hz) correlated positively with expected value estimated from computational modeling during reward-based learning (after outcome delivery) or punishment-based learning (during the choice process). Furthermore, LFO recorded from ATN/DMTN were also negatively correlated with outcomes so that both components of reward prediction errors were signaled in the human thalamus. The observed differences in the prediction signals between rewarding and punishing conditions, shed light on the neural mechanisms underlying action inhibition in punishment avoidance learning. Our results provide direct evidence of the thalamus' role in reinforcement-based decision-making in humans and reveal new insights into the neural computations that underlie this structure.

Main text

Line 29: “would put it” implies that this is not a direct quote, but the quotations marks suggest otherwise. The book from which the quote comes should be cited.

We corrected the sentence and added the corresponding citation in the references.

Introduction Line 37: As the philosopher John Locke put it “*reward and punishment are the only motives to a rational creature: these are the spur and the reins whereby all mankind is set on work and guided*”¹.

References Line 508: Locke, J. *Some Thoughts Concerning Education*. (Churchill, 1693).

Line 35-36: Please list the structures deemed to be involved in this circuit and cite evidence for involvement of the thalamus, including whether the evidence suggests the thalamus as a whole or particular nuclei or subnuclei. What is meant by “encodes variables”?

We expanded this section of the introduction to specify the structures of the circuit and changed “encodes variables” by “associated with subject’s expectations or with the experienced outcomes during learning”:

Introduction Line 42: The striatum receives inputs from both cortical and thalamic regions and is densely innervated by midbrain dopamine neurons. Information is then relayed back to the cortex through the basal ganglia, which project through the thalamus. However, there is no evidence in humans regarding how neural activity in the thalamus - a key node in this circuit - is associated with subject’s expectations or with the experienced outcomes during learning.

Introduction Line 67: The dorsomedial (DMTN) and anterior thalamic nuclei (ATN) have a critical role in the prefronto-striatal-thalamic limbic circuit, facilitating flexible behaviors through the updating of stimulus-action-response associations^{23,24,22,16}. ATN and DMTN are both connected to the network of brain regions involved during reinforcement learning, such as the medial prefrontal, cingular and insular cortices²⁵. Functional magnetic resonance imaging studies in humans consistently showed that thalamic signals correlate with the difference between predicted and actual outcomes^{26,14}. Animal electrophysiological and lesion studies suggest that the ATN and the DMTN could play dissociable functions during decision-making^{27,21}. On the one hand, ATN neuronal activity increases during fear conditioning²⁸ and plays a causal role in aversive avoidance tasks¹⁶ while ATN lesions do not affect the learning of response-reward associations¹⁹. Given its strong connections with the medial prefrontal cortex, there were comparatively more studies in the functions of DMTN during reward-based tasks and instrumental behavior^{17,18,19,22}. Yet, the precise mechanism by which ATN and DMTN modulate neural activity in the prefronto-striato-thalamic loops during reward vs. punishment-based learning processes were never explored so that this study aimed at testing whether oscillatory activity in these structures could be associated with learning signals^{29,30}.

Line 45: What is meant by “the ventral striatum and ventromedial prefrontal cortex represent reward learning signals”? It would be helpful to state what specifically was measured and deemed to represent a learning signal.

We apologize for this lack of specificity: we meant that the neural signals (measured during either fMRI or intracranial studies) were shown to correlate to the prediction errors in the ventral striatum and ventromedial prefrontal cortex. We developed the introduction in that sense to be more specific.

Introduction Line 55: Intriguingly, both fMRI and intracranial signals indicate that the behavioral asymmetry observed between reward-seeking and punishment avoidance is mirrored by a corresponding neural asymmetry. Interestingly, both fMRI and intracranial signals indicate that the behavioral asymmetry observed between reward-seeking and punishment avoidance is mirrored by a corresponding neural asymmetry. Specifically, BOLD signals (fMRI) or cortical broadband gamma activity (iEEG) in the ventromedial prefrontal cortex exhibit significant correlation with reward prediction errors during reward-based learning. Conversely, BOLD and broadband gamma are preferentially associated with punishment prediction errors in the anterior insula^{12,13,14,15}.

Line 49: It would be adequate to say animal studies and save the space used in listing species.

Sure, the text has been changed accordingly.

Line 54: The clinical reason for thalamic electrode implantation in humans should be briefly stated here.

This statement was added to bring more context to the studied cohort.

Introduction Line 85: To address this gap, we conducted a study leveraging rare direct intracranial neural recordings obtained from the human limbic thalamus of eight patients (Table S1) with pharmacoresistant epilepsy who were implanted for deep brain stimulation of the anterior thalamus³¹.

Line 58: “with computational modeling of the learning behavior results” This is the first mention of computational modeling. The approach and its purpose should be stated briefly here.

The computational modeling is now briefly explained and justified.

Introduction Line 90: To achieve this, we fitted a Q-learning model to each patient's behavioral data to estimate trial-wise values of latent variables (option values and prediction errors). More specifically we looked for correlations with the computational model variables at the decision stage (option-specific value expectations) and the outcome stage (prediction errors). By combining intra-thalamic recordings with computational modeling of learning behavior, our study represents the first investigation of time-resolved choice and learning signals in the human thalamus.

Line 60: The term “pharmacoresistant” would be preferable.

We now used the suggested terminology.

Figure 1: How were the electrode locations confirmed post-operatively? Precise placement is likely to differ between the eight patients, as electrode trajectories are planned on an individual basis to avoid major vessels. It would be helpful to show the post-operative electrode localization based on individual structural imaging for at least one patient to demonstrate how the authors confirmed the electrode locations post-operatively.

For each patient, a postoperative CT Scan was co-registered (using the Rosa software, Zimmer Biomet, Valence, FR) with preoperative CT Scan, a Fast Gray Matter Acquisition T1 Inversion Recovery (FGATIR, see Sudhyadhom et al. Neuroimage 2009) 3T MRI sequence which provided a reliable visualization of thalamic structures (Figure R3). Patient-specific segmentation of thalamic nuclei and electrode localization was done with SureTune (Medtronic, Dublin, Ireland). Briefly, this tool allowed us to fit an atlas to the patient MRI (Bardinet et al., 2009) while the neurosurgeon (S. Chabardes) also performed a manual segmentation to further refine anterior thalamus targeting. We added the **Supplementary Figure S1** to demonstrate how we confirmed the electrode locations post-operatively.

Results Line 138: Upon examining the post-operative CT scans images that were co-registered to Fast Gray Matter Acquisition T1 Inversion Recovery³² (FGATIR) 3T MRI images, it was determined that the electrodes consisted of two upper contact pairs positioned within the anterior thalamic nucleus and of the more ventral contact pairs localized within the dorsomedial thalamic nucleus (Fig. S1). Given the absence of significant differences between thalamic nuclei (ATN vs. DMTN) and sides (Left vs. Right, Fig. S2-3), in the following, all the analyses were conducted across all recording sites.

Methods Line 288: Electrodes were implanted through the ATN to ensure its maximal recording, with at least the two most dorsal contacts inside the ATN. As a result, the more ventral-proximal contacts pointed towards the dorsomedial thalamic nuclei (DMTN) located below the ANT along the implantation trajectory. All electrodes' positions were checked intraoperatively using a 3D X-ray image using the O-ARM tool (Medtronic, Dublin, Ireland). To improve targeting accuracy, post-operative 3D CT-scan, was obtained and merged using ROSA software with preoperative CT-scan, T1 and nd Fast Gray Matter Acquisition T1 Inversion Recovery³² (FGATIR) 3T MRI images. Patient-specific segmentation of thalamic nuclei and electrode localization was done with SureTune (Medtronic, Dublin, Ireland). Briefly, this tool allowed us to fit an atlas to the patient MRI⁴⁹ while the neurosurgeon (S. Chabardes) also performed a manual segmentation to further refine anterior thalamus targeting. As a result, the more ventral-proximal contacts pointed towards the dorsomedial thalamic nuclei (DMTN) located below the ANT along the implantation trajectory.

Figure R3 (Supplementary Figure S1 in the manuscript). Anatomical location of thalamic electrodes. Reconstruction of intra-thalamic contact locations using intraoperative X-ray image coordinates, overlaid onto a preoperative FGATIR MRI mask. Representative data from a single patient: the upper contacts are situated within the anterior thalamic nucleus (ATN) as segmented by the neurosurgeon, while the lower contacts are positioned in the dorsomedial thalamic nucleus (DMTN). The ventricular system/third ventricle is depicted in orange, the ATN is shown in blue (right) and green (left), the electrodes are displayed in gray, and their trajectories are represented by dotted green lines. (A) MRI view of the nucleus implantation, aligned with the electrode orientation. (B) 3D reconstruction of the electrode within the nucleus.

Line 78: A brief explanation of what is meant by a Q-learning model is needed here. The authors state that they fitted a Q-learning model, but Q-learning is a model-free reinforcement learning algorithm.

We thank the Reviewer for identifying a lack of clarity concerning our explanation. Part of the ambiguity arises from the fact that, in the field (reinforcement learning), the term ‘model’ indicates both a ‘computational model’ (i.e., a set of logico-mathematical operations that represents a reinforcement learning algorithm - the Q-learning being one example) and ‘environment model’ (i.e., an explicit representation of how states are connected in a given situation - something that is absent in the Q-learning). So, somehow counterintuitively, the Q-learning model is a “model-free model”, because it is a computational model, but it lacks an explicit representation of the environments. This being said, we modified the text in order to make the introduction of these concepts (hopefully) clearer.

Results line 128: To do so, we fitted a Q-learning model (QL) to behavioral data to estimate trial-wise option-specific expected values and prediction errors. The QL model generates choice probabilities applying a SoftMax function to option values (Q-values), which are then updated at the time of outcome via a prediction error minimization process^{2,7}. Fitting the model means adjusting its two parameters (learning rate and choice temperature) to maximize the likelihood of observed choices (see Methods).

Line 79: How were there 48 recording sites? Eight patients each had bilateral implantation of electrodes with 4 contacts each, suggesting 64 recordings sites. Re-referencing first appears at line 241. Presumably the authors meant 48 bipolar channels?

Yes, exactly, the 48 recording sites do correspond to contact-pairs. We clarified this issue in the revised manuscript.

Methods Line 336: LFP signals were recorded with lead extensions connected to an EEG acquisition system (Micromed SD MRI, bandwidth 0.1–200 Hz, sampling rate 1024 or 2048 Hz). Each DBS electrode consisted of 4 contacts with a length of 1.5 mm, separated by 0.5 mm ((deep brain stimulation macro-electrode 3389, Medtronic, Minneapolis, US). Signal processing was performed using a longitudinal bipolar montage between the 3 adjacent pairs of contacts per electrode to maximize the sensitivity to local sources of LFP. Overall, 48 bipolar channels were recorded (3 contact-pairs/electrode × 2 hemispheres × 8 patients) using a commercial video-EEG monitoring system (System Plus, Micromed).

Results Line 136: The neural activity of each recording site (n=48 bipolar channels, see Methods) was then regressed in the time-frequency domain against both expectation and outcome signals at different time points during the task.

Line 81: What is meant by “the absence of significant differences between sites located within the anterior thalamic nucleus (n=16 sites), the dorsomedial thalamic nucleus (n=16 sites, Supplementary. Fig. S1) or sites localized in-between (n=16)”? The anterior and dorsomedial nuclei have different anatomical connectivity, consistent with engagement in different networks. What neural activity measure was used? Were the data baselined?

We rephrase this paragraph to improve its clarity. The neural activity used was the LFO, based on the identified cluster of the time-frequency and control analyses (see **R1.2**). The data were z-scored trial-wise by the power in a 10 seconds epoch centered on the event of interest. This important information was added to the method section.

Results Line 136: The neural activity of each recording site (n=48 bipolar channels, see Methods) was then regressed in the time-frequency domain against both expectation and outcome signals at different time points during the task. Upon examining the post-operative CT scans images that were co-registered to Fast Gray Matter Acquisition T1 Inversion Recovery³² (FGATIR) 3T MRI images, it was determined that the electrodes consisted of two upper contact pairs positioned within the anterior thalamic nucleus and of the more ventral contact pairs localized within the dorsomedial thalamic nucleus (**Fig. S1**). Given the absence of significant differences between thalamic nuclei (ATN vs. DMTN) and sides (Left vs. Right, **Fig. S2-3**), in the following, all the analyses were conducted across all recording sites.

Methods Line 353: Time-frequency power was converted into dB (decimal logarithm transformation) to improve the Gaussian distribution of the data and thereafter baselined using a trial wise z-score transform (using the average power in a 10 s time window centered on the event of interest), as previously described^{13,50}.

Figure 2: Labelled colorbars are needed. Is power shown or T-values? The axes should be extended to show the significant clusters in their entirety. A correction should be applied to take account of recording from 4 structures: left and right ATN and left and right DMTN. What is meant by “significant clusters (cluster-corrected, $p < 0.05$) in the time domain for a one-sample t-test against 0”?

We have included labeled color bars in all the figures of the revised manuscript. We also have clarified that the figures represent two-sided one-sample t-tests against 0 of the regression estimates across recordings sites. Regarding the axes, we have extended them to show the significant clusters in their entirety. We also expanded the statistical test section in the method to clarify what we did. **Figure 2** does not distinguish ATN from DMTN since we pooled the data from the four structures (left and right ATN and left and right DMTN) so that the multiple comparison issue only concerns the time dimension. Our reasoning was that for either RPE or PPE, we did not find any significant differences between the structures, the sides, or the interaction between structures and sides even when no correction for multiple comparison was applied for the side or structure factors (see **R1.1, Figure S2-3**).

We rephrase the statement "significant clusters (cluster-corrected, $p < 0.05$) in the time domain for a one-sample t-test against 0". It referred to the results of our statistical analysis that was based on permutation tests to control for multiple comparisons in the time (or time-frequency) domains, whereby the pairing between power and regressor values across trials was shuffled randomly 60,000 times. We then extracted the maximal cluster-level statistics (the sum of t-values across contiguous time points passing a significance threshold of 0.05) for each shuffle to compute a ‘null’ distribution of effect size. For each significant cluster in the original (non-shuffled) data, we computed the proportion of clusters with higher statistics in the null distribution, which is reported as the ‘cluster-level corrected’ p value.

Figure R4. (Figure 2 in the manuscript). Thalamic low-frequency oscillations associated with choice expectations during choice. a-b. Time-frequency decomposition of expected value signals following cue onset(a) or preceding participant's choice (b). Hotter (cooler) colors indicate more positive (negative) regression estimates (between thalamic power and Q-values). Black contours delimit statistical thresholds from $p < 0.05$ to $p < 5.0 \times 10^{-6}$. Significance was assessed using multiple one-sample student t-tests against zero across all thalamic sites ($n=48$). The boundaries of the frequency bands delta (δ : 1-4 Hz), low-frequency oscillations (LFO: 4-12 Hz), beta (β : 12-33 Hz), and gamma (γ : 50-150Hz) at indicated on the right side. Shaded area represents the standard deviation of the reaction time giving the timing of the choice (a) or go signal onset (b). **c-d.** Time-course of average (solid lines) regression estimates obtained from linear fit of LFO with Q_r or Q_p after the cue onset (c) and before the choice (d). Shaded gray areas around the mean represent SEM across sites. Colored horizontal bars displayed on the top of c-d indicate significance ($p < 0.05$) for one-sample t-tests against 0 in the reward (green) and punishment conditions (red) or for paired t-tests comparing the regression estimates in the reward and punishment conditions. Reaction times (RT) in the reward and punishment conditions are represented as circles (reward: green; punishment: red) and horizontal lines (mean \pm sd).

Methods 420: The significance of regression estimates across thalamic sites was assessed at each time-frequency point using a one-sample two-tailed t-test against zero across all bipolar channels. Statistical significance was assessed through permutation tests as previously. First, the pairing between neural responses and predictors across trials was shuffled randomly 300 times for each recording site. Second, we performed 60,000 random combinations of all sites, drawn from the 300 shuffles calculated previously for each site. The maximal cluster-level statistics (the maximal sum of t-values over contiguous time points exceeding a significance threshold of 0.05) were extracted for each combination to compute a “null” distribution of effect size. The p-value of each cluster in the original (non-shuffled) data was finally obtained by computing the proportion of clusters with higher statistics in the null distribution, and reported as the corrected p value noted (p_c).

Figure 3: Line 185: Please define PE.

Thanks, we actually removed the PE abbreviation.

Line 87: How was Q calculated? A brief explanation would be helpful here.

We rephrase the paragraph about QL to better explain what was done.

Results line 128: We next investigated the association between thalamic neural activity and reinforcement learning variables. To do so, we fitted a Q-learning model (QL) to behavioral data to estimate trial-wise option-specific expected values and prediction errors. The QL model generates choice probabilities applying a SoftMax function to option values (Q-values), which are then updated at the time of outcome via a prediction error minimization process^{2,7}. Fitting the model means adjusting its two parameters (learning rate and choice temperature) to maximize the likelihood of observed choices (see Methods).

Line 132: “LFO oscillations”. Word repetition; O stands for oscillations already.

We deleted the repetition.

Line 139: Typo: “are” rather than “and”?

Right.

Line 140: The authors appear to have tested only the 4-12 Hz frequency band., so it is not clear how they addressed “another open question in the field, which is to test the frequency bands involved during learning”. To address the question as to which frequency bands are involved, direct comparisons should be made between frequency bands.

We thank the reviewer for raising this point. We have directly compared the contribution of LFO to the contribution of all frequency bands to the regression with prediction error in the reward and punishment conditions (see our response to this major comment above: R1.2).

Methods

Line 200: The authors mention pre-operative planning of the electrode locations, but how were the actual locations confirmed post-operatively?

Electrode's locations were confirmed post-operatively by a neurosurgeon using intraoperative X-ray image coordinates overlaid onto a preoperative FGATIR MRI mask (**Figure S1**). We added the Supplementary Figure 1 to demonstrate how we confirmed the electrode locations post-operatively (see our more detailed response page 8 of this document) in response to another minor comment.

Line 212: The authors report that no seizures took place during the testing sessions. Were the data inspected for interictal epileptiform discharges, which could have a potential influence on attention?

Upon receiving this feedback from the reviewer, we conducted a thorough examination of the data to identify trials that potentially involved epileptiform interictal activities and were considered unsuitable for analysis. For each recording site, we employed a sliding window of 50 ms within each epoching window. We then eliminated trials exhibiting a power that sporadically surpassed five times the standard deviation of the average signal. Consequently, an average of 6.1% of trials were excluded from each epoching window. This exclusion rate primarily stems from two patients, who had an average of 20.6% of trials excluded. All the reported results in the revised manuscript were obtained on the cleaned data set so that all statistical analyses and figures have been updated.

Methods Line 353: Time-frequency power was converted into dB (decimal logarithm transformation) to improve the Gaussian distribution of the data and thereafter baselined using a trial wise z-score transform (using the average power in a 10 s time window centered on the event of interest), as previously described^{13,50}. To remove artifacts and pathological interictal epileptiform discharge, we employed the following procedure for each recording site. A sliding window of 50 ms was employed within each event of interest (10 s time window centered on each event). Trials exhibiting a power that sporadically surpassed five times the standard deviation of the average signal were excluded. Consequently, an average of 6.1% of trials per patient were excluded from each epoching window. This exclusion rate primarily stems from two patients, who had an average of 20.6% of trials excluded.

Line 207: "Patients were investigated either in the epilepsy departments of Grenoble, Paris, or Marseille." Paris is not mentioned in Table S1.

Thanks for pointing out this typo. We removed Paris from the main text.

Line 208: Reference should be made to the ethical standards applied. Was the Declaration of Helsinki considered?

Yes, the Declaration of Helsinki was considered.

Method Line 304: Written informed consent was obtained from all participants, and the study was approved by the ethics committee (Comité de Protection des Personnes Sud-Est I, protocol number: 2011-A00083-38) in accordance with the principles of the Declaration of Helsinki.

Report

I was not able to find the data in the figshare repository. Could there be a typo in the link provided?

Thank you for bringing this to our attention. We apologize for any inconvenience caused. We have reviewed the link provided, the correct link to access the data on Figshare is <https://figshare.com/s/a8073058cc36f398e2d1>.

Reviewer #2

The role of the thalamus in human reinforcement learning

In this paper, Collomb-Clerc et al. report on a rare and valuable set of intracranial recordings from the thalamus of intractable epilepsy patients. Patients completed an instrumental learning task in which they had to maximize monetary gains by learning the reward contingencies associated with different abstract visual cues, in both reward (+\$) and punishment (-\$) conditions. The authors use a computational framework (Q-learning) to study the representation of reward signals in electrophysiological activity in two different thalamic subregions. Using a multi-frequency resolved regression approach, the authors propose that low-frequency activity (mostly in the 4-12Hz range) reflects expected values and outcomes at different times in the task, with different encoding profiles for reward and punishment. The authors conclude that thalamic low frequency oscillations encode expectation and reward prediction errors.

This is one of the first papers to relate intracranial signals to RL derived parameters, one of few papers to examine thalamic activity in awake behaving human patients, and the first one to do both at the same time. The choice of task is appropriate, the regression approach adequate (if incomplete – see below) and the electrophysiological methods are equally appropriate. The number of patients is about average for a study of these characteristics. There is great novelty in these thalamic recordings and the methodological combination, which has the potential to shed new light into the involvement of anterior thalamus in reward-guided behavior.

We thank the reviewer for these supportive comments.

R2.1 Despite these strong points, there are multiple aspects of the paper that greatly diminished enthusiasm. First, behavioral performance for most of the patients in the dataset is borderline, with many of the patients performing barely above chance in either condition (Fig. 1d). This throws into question to what extent the patients really understood the task, which in turns has serious implications for the computational models being used in the analysis. Some more strict inclusion criteria may be appropriate, or individual-level modeling that may indicate that the chosen Q-learning model does a better job of explaining the behavioral strategy than a random 50/50 model or heuristic strategies.

We appreciate this important point concerning behavioral performance. Before providing some additional analyses, let us explain that we respectfully disagree with the Reviewer's impression that performance was "borderline" in our sample. Indeed, correct choice rate in our cohort of epileptic patients significantly differed from chance in both learning conditions (reward: 65 ± 0.04 , $t(7) = 4.23$, $p = 0.0039$; punishment: 60 ± 0.02 , $t(7) = 5.13$, $p = 0.0014$). Considering the clinical condition of pharmacoresistant epileptic patients, the observed rates of correct choices in our study were surprisingly high when compared to similar studies in the field, such as those conducted with the same (or similar) task in other clinical cohorts, such as Parkinson (~60%; Palminteri et al. Cortex, 2013), Tourette (~63%; Worbe et al, Archives, 2011) and, more recently, epileptic patients (~70%; Gueguen et al., NC, 2021), to name a few.

We believe that our results are even more convincing if we take into account that the above-mentioned scores (65 ± 0.04 and 60 ± 0.02) refer to averages calculated across the whole learning curve, including the first trials where the participant has no or very little clue concerning the options values. Thus, to further check how participants understood the task, we examined the last four trials of each cue pair in every session, i.e., when the cue-action-reward association is presumed to be most effectively learned. This analysis (Figure R5) further confirmed that in both the reward and punishment conditions, the accuracy approaches 70 % and surpassed chance levels (reward: 71 ± 0.06 , $t(7) = 3.78$, $p = 0.0069$; punishment: 68 ± 0.04 , $t(7) = 4.96$, $p = 0.0016$).

Finally, one should keep in mind that, given the stochastic nature of our task (probabilistic outcomes) and cognitive hypothesis (SoftMax decision rule, delta rule-based value learning) it is still quite conceivable that low (or chance-level) performance arises even in a subject actively engaged in the task. For this, an interesting, alternative measure is to check whether or not our subjects (or how many subjects) were better explained by our (minimal) computational model, compared to random responding. To do so, we compared the goodness of fit of the model compared to that of random responding. We used the Akaike Information Criterion (AIC), which is a well-validated metric of model comparison. Even accounting for the extra 2 free parameters (temperature and learning rate - while the random "model" has zero free parameters), the QL learning model displayed a significantly lower AIC (indicating better fit): 642 ± 40 vs. 732 ± 36 ($t(7) = -3.2$, $p = 0.015$). The log Likelihood was smaller for the QL

model than for random response for all patients, and the average log Likelihood of patients was significantly lower for the QL model than for random response ($t(7) = -3.20$, p -value = 0.015; **Figure R5**).

Figure R5 (Figure 1 in manuscript). Reinforcement-learning paradigm and behavior. *a.* Schematic figure of the position of the deep brain stimulation electrodes used to record intra-thalamic signals (ATN: anterior thalamic nucleus; DMTN: dorsomedial thalamic nucleus; TH: Thalamus; HTH: Hypothalamus; GPi/GPe: Globus pallidus intern/extern; LV: Left ventricle). *b.* Successive screenshots of a typical trial in the reward (top) and punishment (bottom) conditions. Patients had to select one abstract visual cue among the two presented on each side of a central visual fixation cross and subsequently observed the outcome. Durations are given in seconds. *c.* Average \pm SEM reaction times across patients ($n = 8$) through trials shown separately for the reward (Rew, green) and punishment (Pun, red) conditions. *d.* Average \pm SEM reaction times across patients in the reward and punishment conditions. Dots represent data from individual patients. Asterisk indicates the significance of a paired t -test comparing reaction times between conditions. *e.* Average \pm SEM choice performance across patients. The average predicted performance from a fitted Q -learning model is indicated by a white dot for each condition. Gray dots represent data from individual patients. Asterisk indicates the significance of the one-sample t -test used to compare for each condition the correct choice rate to the chance level (i.e., 50%). *f.* Average \pm SEM learning curves across patients ($n = 8$) through trials shown separately for the reward and punishment conditions. *g.* Average \pm SEM choice performance across patients restricted to the last 4 trials of each condition. *h.* Average Akaike Information Criterion (AIC) of Q -learning (QL) model versus random choices across patients. Dots represent data from individual patients. Asterisk indicates the significance of the two-tailed paired t -test used to compare the AIC of the QL model versus random choices.

We modified Figure 1 to integrate these new results and updated the result section to integrate these changes as follows:

Results Line 107: Behavioral results were consistent with what was previously observed in this task (**Fig. 1**). Reaction times (**Fig. 1c-d**) were significantly shorter in the reward (1173 ± 164 ms) than in the punishment (1726 ± 291 ms) condition ($t(7) = -3.10$, $p = 0.017$). Accuracy was on average (**Fig. 1e-f**) higher than chance in both the reward (65 ± 0.04 , $t(7) = 4.23$, $p = 0.0039$) and punishment conditions (0.60 ± 0.02 , $t(7) = 5.13$, $p = 0.0014$) and was not different between the two conditions ($t(7) = 1.68$, $p = 0.14$). To further check how well participants understood the task, we next examined the last four trials of each cue pair in every session, i.e., when the cue-action-reward association is presumed to be most effectively learned. This analysis (**Fig. 1g**) confirmed that in both the reward and punishment conditions, the accuracy approaches

70 % and surpassed chance levels (reward: 0.71 ± 0.06 , $t_{(7)} = 3.78$, $p = 0.0069$; punishment: 0.68 ± 0.04 , $t_{(7)} = 4.96$, $p = 0.0016$; two-tailed paired student t-test). Finally, to assess whether or not our participants were better explained by a QL model compared to random responding, we compared the goodness of fit of the QL model compared to that of random responding using the Aikake Information Criterion (AIC). This confirmed that the QL model displayed a significantly lower AIC (indicating better fit, see Fig. 1h; random responding: 642 ± 40 ; QL: 732 ± 36 ; $t_{(7)} = -3.2$, $p = 0.015$; two-tailed paired student t-test). Thus, patients learned similarly from rewards and punishments but took longer to choose between cues for punishment avoidance, in line with previous behavioral data from healthy subjects⁷ or epileptic patients¹². These results confirm that, although instrumental performances are similar, the decision process differs in reward-seeking and punishment-avoidance contexts in a way that is compatible with a motor inhibition induced by punishment expectation^{8,9,10}.

R2.2. The second major point relates to the relationship between neural activity and computational parameters (Figs.2 and 3). There are many concerns with these results (see below), but the overall point is that the specificity of the results, and whether the activity patterns presented correspond to value computations or to more unspecific responses. There are multiple reasons for this reservation. First, the very sustained activations (primarily Fig. 2a-b) are suspect. For example, activation related to Q-values is sustained for over 3s (Fig. 2a), which is a very long time and not in line with similar value-related activations elsewhere (to use an example from the author's see Lopez-Persem et al 2020) which are noticeably shorter. It is of course possible that the timecourse of thalamic activation is very different from cortical ones, but this may be unlikely given that cortex is presumably driving this activity.

We thank the reviewer for pointing this out. We think that the new versions of figure 2-3 and the following responses to reviewer 2 will address all the concerns listed.

At the conceptual level, although it is interesting to compare the temporal dynamics between (1) expected (learned) values in the thalamus examined during reinforcement learning in this study and (2) value (experienced) in the cortex examined during a valuation task in our previous study (Lopez-Persem et al., Nature Neuro 2020), the discrepant temporal dynamics between studies could be driven by many factors such as the task (pleasantness ratings vs. reinforcement learning) or the neural proxy used (broadband gamma vs. LFO). Nevertheless, we had the opportunity to overcome both of these limitations by re-analysing the data from Gueguen et al., (Nat. Comm. 2021) in which epileptic patients explored with depth electrodes sampling the cortex performed an identical learning task; we extracted LFO in hippocampus, ventromedial, dorsolateral and orbitofrontal cortices because these regions were previously shown to be associated with prediction error signals.

We found that the timing and amplitude of the thalamic regression estimates between LFO and prediction errors are remarkably consistent with the regression estimates observed in from hippocampus, lateral orbitofrontal cortex, dorsolateral prefrontal cortex, and ventromedial prefrontal cortex (see below, **Figure R6**). The main qualitative difference is that in several cortical areas, there was an initial positive association between LFP and prediction errors which then reverted to a negative association (i.e., sign reversals) whereas this pattern was absent in the thalamus. That said, the timing of the neural response to prediction errors is globally consistent, with a comparably long return to baseline in the vmPFC, IOFC dIPFC and the thalamus, for instance. This suggests that the sustained activity is an inherent property to the LFO during this task.

We integrated this re-analysis of the data from Gueguen et al. 2021 in the result and discussion sections of the manuscript. We thank the reviewer for this excellent point.

Results Line 200. To compare the time courses of the association between thalamic and cortical LFO and prediction errors, we re-analyzed a data-set¹³ in which we recorded intracerebral data from the hippocampus, orbitofrontal and prefrontal regions during an identical task. We found that the temporal dynamics of LFO associated with prediction errors were similar between thalamic and cortical sites (**Fig. S4**). The main qualitative difference was that in hippocampal and cortical areas, there was an initial positive association between LFO and prediction errors which then reverted to a negative association (i.e., sign reversals) whereas this initial increase was absent in the thalamus.

Discussion line 240: The thalamic reward prediction error signals likely reflect a cortical input from the ventromedial prefrontal cortex / lateral orbitofrontal cortex which also exhibited sustained LFO modulations (lasting about 2.5 s), compared to the faster (<1.5 s) dynamics observed when using broadband gamma activity as a neural proxy during value rating⁴² or during reinforcement learning¹³. This suggests

that reinforcement learning processes trigger a sustained neural activity in the cortico-thalamic circuit involved to implement the teaching signal during reward-based learning.

Figure R6 was added to the manuscript as Supplementary Figure S4.

Figure R6 (Supplementary Figure S4 in manuscript). Comparison of LFO association with prediction errors in the thalamus ($n=8$ patients) and in the cortex ($n=19$ patients from Gueguen et al., 2021). Time-course regression in the 4-12 Hz frequency range with prediction error in the reward (RPE, green) and punishment (PPE, red) with regression estimate averaged \pm SEM (shaded gray area around the mean) across recording sites plotted separately for thalamus ($n = 48$ sites, data from this study), hippocampus (HPC, $n=67$), lateral orbitofrontal cortex (IOFC, $n=140$), and ventromedial prefrontal cortex (vmPFC, $n=54$) (re-analysis of cortical data from Guengen et al. 2022).

R2.3. Even taken at face value, these sustained activations raise many concerns, including the fact that cross-trial correlation in Q values may bleed through across trials. The activation has not died down (and in fact looks close to maximal levels) towards the end of the window in 2b – bleeding well into the following trial given the experimental design (Fig.1). How much longer does this last? Is there an effect of trial t onto $t+1$ activity or regression results? This is especially concerning given that Q_t and Q_{t+1} are likely highly correlated across consecutive trials as the model learns values over time. Secondly, it is possible that the computational time-courses are driven not by the specific computations presented but by coarser value signals. In figs. 2a and 2b, the regression incorporates both punishment and reward blocks. I'm not convinced that is appropriate given the differences in the time course of the regression estimates (Fig.1c and d), and it raises the possibility that simple win vs loss expectations (or contextual signal) could drive the results. Perhaps not a very strong possibility given that patients don't do well in the task, but I would still suggest showing separate regressions for reward and punishment conditions.

We apologize for the lack of clarity regarding the timing of the activations of **Figure 2**. We re-worked the figure and analyses to take into account this important comment:

- We first expanded the time windows in all figures of the manuscript to encompass all significant regression periods, ensuring a more comprehensive representation of the data.
- We also modified figure 2 and figure 3 to systematically separate reward and punishment conditions (such that win/loss expectations can no longer be related to the results).

Regarding the possible bleeding across trials, this explanation does not appear plausible because the experimental conditions were interleaved and participants had to learn four pairs of cues (2 pairs in the reward and 2 pairs in the punishment conditions). Thus, Q_t and Q_{t+1} had 75 % chances to belong to a different cue. In addition, as noticed by reviewer 2, we could not directly test whether $Q_{(t-1)}$ was also associated with LFO because the correlation with Q_t was too high such that it was not possible to directly compare within the same GLM both metrics.

Furthermore, we notice that **Figure 2** does not encompass the next trial as it represents the windows time locked on the cue and choice onsets. To improve figure clarity, we now display shaded areas corresponding to the reaction times (2a) or to the stimulus onsets (2b) or to the next trial onset (**Figure 3**).

Figure R7. (Figure 2 in the manuscript). Thalamic low-frequency oscillations associated with choice expectations during choice. a-b. Time-frequency decomposition of expected value signals following cue onset(a) or preceding participant's choice (b). Hotter (cooler) colors indicate more positive (negative) regression estimates (between

thalamic power and Q -values). Black contours delimit statistical thresholds from $pc < 0.05$ to $pc < 5.0 \times 10^{-6}$. Significance was assessed using multiple one-sample student t -tests against zero across all thalamic sites ($n=48$). The boundaries of the frequency bands delta (δ : 1-4 Hz), low-frequency oscillations (LFO: 4-12 Hz), beta (β : 12-33 Hz), and gamma (γ : 50-150Hz) at indicated on the right side. Shaded area represents the standard deviation of the reaction time giving the timing of the choice or go signal onset (b). **c-d.** Time-course of average (solid lines) regression estimates obtained from linear fit of LFO with Q_r or Q_p after the cue onset (c) and before the choice (d). Shaded gray areas around the mean represent SEM across sites. Colored horizontal bars displayed on the top of c-d indicate significance ($pc < 0.05$) for one-sample t -tests against 0 in the reward (green) and punishment conditions (red) or for paired t -tests comparing the regression estimates in the reward and punishment conditions. Reaction times (RT) in the reward and punishment conditions are represented as circles (reward: green; punishment: red) and horizontal lines (mean \pm sd).

Figure R8. (Figure 3 in the manuscript). Thalamic low-frequency oscillations associated with prediction error components. **a-b.** Time-frequency decomposition of prediction errors expectation (a: Q_r or Q_p) and outcome (b: R or P) components. Hotter (cooler) colors indicate more positive (negative) regression estimates. Black contours delimit statistical thresholds from $pc < 0.05$ to $pc < 5.0 \times 10^{-6}$. Significance was assessed using multiple one-sample student t -tests against zero across all thalamic sites ($n=48$). The boundaries of the frequency bands delta (δ : 1-4 Hz), low-frequency oscillations (LFO: 4-12 Hz), beta (β : 12-33 Hz), and gamma (γ : 50-150Hz) at indicated on the right side. Grey shaded rectangles on the right side of all panels represent the standard deviation of the next cue pair (trial $t+1$). **c-d.** Time-course of average (solid lines) regression estimates obtained from linear fit of LFO with prediction error components (Q_r , Q_p , R, P). Shaded gray areas around the mean represent SEM across sites. Colored horizontal bars displayed on the top of c-d indicate significance ($pc < 0.05$) for one-sample t -tests against 0 in the reward (green) and

punishment conditions (red) or for paired t-tests comparing the regression estimates in the reward and punishment conditions.

R2.4. An even simpler potential explanation that should be formally ruled out is that the activations shown are simply motor or visual. Rather than reflecting value-based quantities, these activations could simply be stimulus-evoked activity (e.g. visual), something that has been previously reported in human thalamus (<https://www.nature.com/articles/s41598-021-96588-x>). Even though the activations are regression results and not simple time-locked power modulations, depending on exactly how the Q-values and outcomes regressors look this could be a seriously confounding issue. This could be potentially addressed by having dummy regressors for confounds (motor, visual) in the regression model in addition to the regressors of interest. Similarly, related to the win/loss possibility – if this was driven by win/loss encoding (which is known to engage low f in humans, e.g. Marco-Pallares et al 2008) rather than parametric Q_v or RPE encoding, a regression model that includes wins/losses in addition to the Q values would partially address this.

We thank reviewer 2 again for pointing this excellent paper to us. Yet, we believe that there are several reasons that make it not possible that motor or visual activities could explain the reported associations between LFO and the learning signals in our study:

- by design, expected values were not correlated with visual stimuli or motor commands (e.g., stimuli-to-condition attribution was randomized across subjects and the mapping between choice of the stimuli and motor command was counterbalanced on a trial-by-trial basis so that no fixed mapping could be established between stimuli, values, and motor responses).
- In fact since the task is also designed to match the visual and motor aspects of reward and punishment learning, the differences between Q_r and Q_p shown in figure 2 ($Q_p > Q_r$, $p < 0.05$) cannot be explained by such factors (notice for instance that options were randomly presented on the left or right side of the screen so that chosen value and motor commands were orthogonal)
- as stated above in response to R2.2, now that we systematically separate reward and punishment conditions in all analyses, the win/loss explanation no longer holds
- after carefully reading the relevant paper pointed by reviewer 2 (Leszczynski et al., 2021), it seems that neither the rapid timing (around 200 ms), nor the sign of the power modulation (an increase of power was observed during about 500 ms after visual stimulus onset) observed in their study seem compatible with our results (see exploratory analyses explained below).

We acknowledge that since we did not record eye movements in the context of this experiment, an open question for future research will be to examine how visual fixations and learning signals may interact with thalamic LFO.

In an attempt to further convince reviewer 2 that low-level (motor or visual) factors can hardly account for our results, we ran an exploratory analysis in which we compared the amplitude of thalamic LFO during trials with a low or high level of expectation (median split) separately for rewarding or punishing outcomes (1€ or -1 €) so that in such analysis the visual stimuli were identical between high vs. low level of expectation (and there was no motor response at outcome onsets). We performed a two tailed paired t-test to compare LFO power between high and low expectation trials in the reward and punishment conditions. This analysis confirmed a significant difference between power in high vs. low expectation trials in the rewarding trials ($t_{47} = -3.16$, $p\text{-value} = 0.0028$) but not in the punishing conditions ($t_{47} = 0.38$, $p\text{-value} = 0.71$), confirming the correlation results of **Figure 3**. This is another argument against the idea that visual stimuli or motor commands could influence our results, since here, the visual stimulus was kept constant. It also shows that rewards or punishments induce a decreased level of LFO in contrast to previously reported increased level of LFO using natural visual stimuli in previous studies. This difference speaks in favor of radically different mechanisms modulating differentially the neural activity within the ATN/DMTN according to the task context (memory, natural vision or decision-making tasks).

Figure R9 (Supplementary Figure S5 in manuscript). Average across thalamic LFO site at outcome onset (0-2 s). **a.** Average across thalamic sites (\pm SEM) LFO power in the 0-2s window following the outcome for each outcome in reward (green) and punishment (red) conditions. A two tailed paired t-test was performed to compare LFO power in the reward and punishment conditions revealed a significant difference between outcomes in both the reward condition ($t_{(47)} = 4.77$, $p\text{-value} = 1.85e-05$) and the punishment condition ($t_{(47)} = 6.3161$, $p\text{-value} = 8.93e-08$). **b.** Post-outcome decrease of LFO power is modulated by the level of expectation in the reward condition (green, median split on Q_r) but not in the punishment condition (red, median split on Q_p). Average across thalamic sites (\pm SEM) LFO power in the 0-2s window following the outcome onset separately estimated for low or high expectation trials (median split done separately for each subject and separately for the reward and punishment condition). A two tailed paired t-test was performed to compare LFO power between high and low expectation trials in the reward and punishment conditions revealed a significant difference between power in high vs. low expectation trials in the rewarding trials ($t_{(47)} = -3.16$, $p\text{-value} = 0.0028$) but not in the punishing conditions ($t_{(47)} = 0.38$, $p\text{-value} = 0.71$).

R2.5. There are other aspects of the data that seem to contradict the authors' interpretation that these activations reflect value. The clearest one is presented in Fig. 3d (and Fig. S2): reward and punishment regression estimates look basically identical in sign, magnitude and timecourse. If this was a value signal, wouldn't one expect neural responses to rewards and punishments to be opposite in sign? Again, these responses look very unspecific and generate significant doubts into the reward interpretation.

The reviewer here raises another important point. Indeed, positive outcomes (rewards) and negative ones (punishments) were encoded with the same directionality in LFO (Figure 3 and 4). The question arises whether these signals therefore correspond to the *value* of the outcomes or to their behavioral *saliency* (after all, both rewards and punishments are more salient compared to the neutral outcome). This is not an easy question to answer (even though it is an important one to acknowledge; see our modifications below). The difficulty of answering this question comes from the intrinsic complexity of interpreting directionality of correlation when dealing with LFO power analysis, but also more broadly from considerations of the organization of opponent brain circuits for reward and punishment learning. To address this point, we believe that is useful to take a step back and look what outcome related signals looked like in cortical brain areas traditionally associated with rather reward learning (vmPFC, OFC) or punishment learning (Insula, dlPFC).

Figure R10. (Illustration from Gueguen et al., 2021). Anatomical dissociation of reward vs. punishment prediction errors using broadband gamma activity as a neural proxy in the cortex.

As it can be seen in the **Figure R10** (taken from Gueguen et al.), while it is true that in “positive areas” (IOFC, vmPFC) encoded more robustly rewards and “negative areas” encoded more robustly punishments (dlPFC, aINS) in each set of areas both rewards and punishments were encoded with deviations on the same direction (positive in this case). This was true also for areas that, for instance at the BOLD signal-level are traditionally reported to encode outcomes on a linear scale. This raises the possibility that there is something specific to the LFO that makes inference concerning the directionality of the effect (and its relation with other form of physiological signals) interpreted with cautiousness.

The second point concerns the supposed (and quite well validated) organization of reward and punishment networks into opponent systems (see **Figure R11**, taken from Palminteri et al. 2017) and the specific role of the thalamus as a “hub” for basal ganglia inputs to the cortex.

Figure R11 (Palmiteri et al., 2017) Various hypotheses concerning the neural implementation of punishment avoidance (in red), as opposed to reward seeking (in green). For each hypothesis, the key regions and connections of each opponent system are shown on the left, with their theoretical pattern of activity as a function of prediction error (PE) plotted on the right.

The similar direction of the encoding of the rewards and punishments in the thalamus may reflect the fact that this structure is playing a similar role for otherwise opponent areas (e.g., the vmPFC and the dlPFC) in both conditions. Future studies, involving simultaneous recording of the thalamus and other cortical (“positive and negative”) areas could help shed light on this aspect.

To conclude, we added a caution note on this particular result to discuss the possible involvement of thalamic LFO in the signaling of surprise/saliency signals.

Discussion Line 235: Conversely, the association between thalamic LFO and outcomes (rewards and punishments) went in the same direction in both learning conditions. The similar directionality of outcome encoding may prima facie suggest that thalamic LFO signals behavioral saliency. However, we also note the positive association with reward expected value at the moment of choice and the moment of outcome (i.e., prediction error encoding) suggests that the expected reward value is encoded in thalamic LFO.

R2.6. A relatively minor point – the regression estimates in Fig. 1c and 1d seem pretty tight (low SEM) given that these analyses include all the electrodes in the sample. I would be curious to know what the differences in encoding is across patients (esp. in light of behavioral variability) and electrodes. I would be surprised if these activations were so robust that they appeared in every thalamic electrode, but it would be good to know either way.

This is an excellent point. We run the analysis suggested by the Reviewer and we found that electrophysiological results were quite robust and stable across subjects (with negative association with predictions errors clearly visible in 7 out of 8 subjects (**Figure R12, Supplementary Figure S6 in the manuscript**), with some variability across recording sites which is expected in any biological study.

Figure R12 (Supplementary Figure S6 in manuscript) Individual electrophysiological data. Individual average \pm SEM across recording sites of regression estimates of prediction error against LFO power in the 0-2s window following the outcome in the reward and punishment conditions. Dots represent each recording site ($n=6$ per patient and condition), black (grey) dots corresponding to a recording site with a (un)significant regression.

Results Line 181. We also assess how reliable were the associations between thalamic LFO and prediction errors; we found that 7 out of 8 patients displayed a significant negative association between LFO and prediction errors (**Fig. S6**).

R2.7. In a different but related line, it would be very valuable to look at the time frequency representations (raw power, without the computational analysis) to see if there are power modulations at the time where the computational associations may be happening.

We do agree that showing the raw power would provide additional information. Since we now clearly demonstrated that the relevant information was in the LFO, we looked directly at the LFO across the four types of outcomes (reward: 0\$ vs. +1\$; punishment -1\$ vs 0\$). This nicely complemented what we showed in figure 3 using the correlation analysis for the outcome component of the prediction errors. By the way, notice that reward (R: 1 vs 0 €) and punishments (P: -1 vs. 0 €) are already model-free variables in figure 3. The negative correlation between LFO and R or P was driven by a lower LFO power for 1€ (wins or losses) relative to neutral outcomes (0 €). A two tailed paired t-test was performed to compare LFO power in the reward and punishment conditions revealed a significant difference between outcomes in both the reward condition ($t_{47} = 4.77$, $p\text{-value} = 1.85e-05$) and the punishment condition ($t_{47} = 6.3161$, $p\text{-value} = 8.93e-08$). We added a supplementary Figure 5 and a brief comment in the result section to document this issue.

Results Line 172: Both types of outcomes were significantly encoded by LFO in both rewarding and punishing conditions (**Fig. 3d**; 0.39 to 1.95 s window, $\beta_R = -0.21 \pm 0.01$, $\text{sum}(t_{(47)}) = 113.56$, $p_c < 0.05$; 0.63 to 1.80 s window, $\beta_P = -0.20 \pm 0.01$, $\text{sum}(t_{(47)}) = 83.99$, $p_c < 0.05$). These negative associations between LFO and outcomes were driven by stronger decrease of LFO when winning or losing money compared to neutral outcomes (**Fig. S5**).

Figure R13 (Supplementary Figure S5 in manuscript). Average across thalamic LFO site at outcome onset (0-2 s). **a.** Average across thalamic sites (\pm SEM) LFO power in the 0-2s window following the outcome for each outcome in reward (green) and punishment (red) conditions. A two tailed paired t-test was performed to compare LFO power in the reward and punishment conditions revealed a significant difference between outcomes in both the reward condition ($t_{(47)} = 4.77$, $p\text{-value} = 1.85e-05$) and the punishment condition ($t_{(47)} = 6.3161$, $p\text{-value} = 8.93e-08$). **b.** Post-outcome decrease of LFO power is modulated by the level of expectation in the reward condition (green, median split on Q_r) but not in the punishment condition (red, median split on Q_p). Average across thalamic sites (\pm SEM) LFO power in the 0-2s window following the outcome onset separately estimated for low or high expectation trials (median split done separately for each subject and separately for the reward and punishment condition). A two tailed paired t-test was performed to compare LFO power between high and low expectation trials in the reward and punishment conditions revealed a significant difference between power in high vs. low expectation trials in the rewarding trials ($t_{(47)} = -3.16$, $p\text{-value} = 0.0028$) but not in the punishing conditions ($t_{(47)} = 0.38$, $p\text{-value} = 0.71$).

R2.8. Finally, the authors report no activation in relatively higher frequencies, contrary to what has been described in cortical recordings. I realize that because of sampling rate limitations it would be difficult to try to extend analysis into the high gamma band, and the authors cautiously (and I think correctly) limit their analyses to <128Hz, but even that should be able to capture gamma and some broadband high gamma activations. This is surprising enough that it merits some discussion - do they have an interpretation for this?

This is another excellent point (the sampling rate varied between 512 Hz to 2048 Hz in this study). Therefore, following R1 and R2 suggestions, we now provide separate regressions with RPE or PPE across four delta (1-4 Hz), LFO (4-12 Hz), beta (13-33 Hz), and gamma (50-150 Hz) frequency bands. More precisely, for each frequency band and recording site, power time series were averaged over the outcome time window (0 - 2 s) and regressed against PE separately for the reward and punishment conditions. This analysis revealed that LFO were significantly associated to PE encoding in the reward ($t_{(47)}=-4.15$, $p=0.00014$) and the punishment conditions ($t_{(47)}=-5.73$, $p=6.89e-07$). The other frequency bands did not exhibit any significant association with PE neither in the reward (delta: $t_{(47)}=1.0$, $p=0.32$; beta: $t_{(47)}=-0.039$, $p=0.97$; gamma: $t_{(47)}=-0.61$, $p=0.54$) nor in the punishment condition (beta: $t_{(47)}=1.24$, $p=0.22$; gamma: $t_{(47)}=-0.27$, $p=0.79$) at the exception of the delta band in the punishment condition (delta: $t_{(47)}=2.08$, $p=0.042$).

Furthermore, to quantify the respective contribution of activities in the different frequency bands to PE signaling across thalamic sites, we next included them as separate regressors in general linear models meant to explain PE. To further check whether any frequency band could provide additional information about PE, we fitted separately RPE or PPE with all possible general linear models (GLMs) containing 4-12 Hz together with any combination of other frequency bands (see Materials and methods). Bayesian model selection designated the 4-12 Hz-only GLM as providing the best account of both types of prediction errors (RPE: $E_f = 0.9821$, $X_p = 1$; PPE: $E_f = 0.9821$, $X_p = 1$). Thus, even if delta-frequency activity was significantly related to PPE, it carried redundant information relative to that extracted from 4-12 Hz.

We added Figure R2 as a supplementary figure to the manuscript to illustrate these new analyses. We also added specific sections within the result and method parts of the manuscript as follows:

Figure R14 (Figure 4 in the manuscript): Contribution of frequency bands to prediction error encoding in the punishment (red) and reward (green) conditions. Average across sites of the regression estimates of power against prediction errors for the frequency bands delta (1–4 Hz), LFO (4–12 Hz), beta (13–33 Hz) and gamma (50–150 Hz). LFO power was averaged over 0–2 s post-outcome window. Stars indicate significance ($p < 0.05$) of regression estimates (one-sample, two-sided Student’s t-test). Dots correspond to regression estimates across trials for each recording site ($n = 48$). RPE: reward prediction error. PPE: punishment prediction error.

Results Line 185: To ensure that our focus on LFO was justified, we explored activity in other frequency bands (Fig. 4). This analysis revealed that LFO were significantly associated to prediction errors in the reward ($t_{(47)} = -4.15$, $p = 0.00014$) and the punishment conditions ($t_{(47)} = -5.73$, $p = 6.89e-07$). The other frequency bands did not exhibit any significant association with prediction errors neither in the reward (delta: $t_{(47)} = 1.0$, $p = 0.32$; beta: $t_{(47)} = -0.039$, $p = 0.97$; gamma: $t_{(47)} = -0.61$, $p = 0.54$) nor in the punishment condition (beta: $t_{(47)} = 1.24$, $p = 0.22$; gamma: $t_{(47)} = -0.27$, $p = 0.79$) at the exception of the delta band in the punishment condition (delta: $t_{(47)} = 2.08$, $p = 0.042$). To further check whether any frequency band could provide additional information about prediction errors, we next fitted separately reward or punishment prediction errors with all possible general linear models (GLMs) containing LFO together with any combination of other frequency bands (see Methods). Bayesian model selection designated the LFO-only GLM as providing the best account of both types of prediction errors (RPE: $E_f = 0.9821$, $X_p = 1$; PPE: $E_f = 0.9821$, $X_p = 1$). Thus, even if delta-frequency activity was significantly related to prediction errors in the punishment condition, it carried redundant information relative to that extracted from LFO.

Methods Line 363:

Contributions of frequency bands. To assess the contribution of the different frequency bands to prediction error representation, reward prediction errors (RPE) or Punishment prediction errors (PPE) were regressed separately across trials against power P (normalized envelope) of each frequency band, averaged over time between 0 and 2 s after outcome onset:

$$RPE = \alpha + \beta P + \epsilon$$

with α corresponding to the intercept and ϵ to the error term. The significance of the regression estimates β was assessed across recording sites using two-sided, one-sample, Student’s t-tests.

In order to determine whether other frequency bands provided additional information relative to LFO, the following 8 GLMs were compared:

$$\text{RPE} = \beta_{\text{LFO}} \times P(\text{LFO}) + \beta_{\delta} \times P(\delta) + \beta_{\beta} \times P(\beta) + \beta_{\gamma} \times P(\gamma)$$

Here, β_{LFO} , β_{δ} , β_{β} and β_{γ} correspond to the regression estimates of the power P in the LFO (4-12 Hz), delta (1-4 Hz), beta (13-33 Hz), and gamma (50-150 Hz) bands in the 0-2 s time-window after the outcome onset. We compared this general linear model with eight possible alternative models:

$$\text{RPE} = \beta_{\text{LFO}} \times P(\text{LFO})$$

$$\text{RPE} = \beta_{\text{LFO}} \times P(\text{LFO}) + \beta_{\delta} \times P(\delta)$$

$$\text{RPE} = \beta_{\text{LFO}} \times P(\text{LFO}) + \beta_{\beta} \times P(\beta)$$

$$\text{RPE} = \beta_{\text{LFO}} \times P(\text{LFO}) + \beta_{\gamma} \times P(\gamma)$$

$$\text{RPE} = \beta_{\text{LFO}} \times P(\text{LFO}) + \beta_{\delta} \times P(\delta) + \beta_{\beta} \times P(\beta)$$

$$\text{RPE} = \beta_{\text{LFO}} \times P(\text{LFO}) + \beta_{\delta} \times P(\delta) + \beta_{\gamma} \times P(\gamma)$$

$$\text{RPE} = \beta_{\text{LFO}} \times P(\text{LFO}) + \beta_{\beta} \times P(\beta) + \beta_{\gamma} \times P(\gamma)$$

$$\text{RPE} = \beta_{\text{LFO}} \times P(\text{LFO}) + \beta_{\delta} \times P(\delta) + \beta_{\beta} \times P(\beta) + \beta_{\gamma} \times P(\gamma)$$

The model comparison was conducted using the Variational Bayesian Analysis (VBA) toolbox⁵¹. Log-model evidence obtained in each recording site was taken to a group-level, random-effect, Bayesian model selection (RFX-BMS) procedure⁵². The RFX-BMS provided an exceedance probability (X_p) that measures the likelihood of a given model being more frequently implemented relative to all the others considered in the model space in the population from which samples are drawn.

R2.9. The discussion section was very short and lacking in detail on the implications of the findings. Very little space is devoted to discussing the proposed notion that low frequency oscillations encode reward computations. This is a pretty novel idea, and a departure from the more conventional role that high frequency oscillations encode local computations for which significantly more evidence exists, and so I would like to see a direct discussion of the potential neurobiological implications, relationship to high frequency activity, etc. More importantly – taken at face value, the results implicate that there are separable bases for expected value and outcome value computations, but how these are combined to give rise to RPE signals (Fig.2) is ignored (a related point – RPEs [Fig. 2a] and outcomes [Fig. 3b] look almost identical, which relates to my point above about specificity). If that is correct, one could expect some systematic relationship between the neural activity underlying Q-values (Fig 3a) and outcomes (Fig 3b) that mirrors (or not) their algorithmic relationship (equation 2), but whether this is the case is not discussed at all. This mechanistic type of insight is where this approach has the potential to really advance the field – not devoting time to this seems like a missed opportunity.

We thank the reviewer for pointing this out. As noted elsewhere the paper was rather short because it was originally aimed to match a “short communication” for Nature Neuroscience. In the revised manuscript, we have now entirely rewritten the Discussion. We have done so in many respects (see previous points in response to both Reviewers), but in particular, this is how we expanded our paragraph concerning the neurobiological implications of our findings:

Discussion Line 252: We observed that reward prediction error was represented in the low-frequency oscillations (4-12 Hz) in the human ATN at the time of outcome onset, but this was not true for higher frequencies. In mice, beta (13-30 Hz) synchrony between the mediodorsal thalamus and the prefrontal cortex was associated with learning¹⁷, whereas in humans, intracranial recording revealed that broadband gamma activity (50-150 Hz) recorded in the cortex encoded reward and punishment-based learning signals¹³. We speculate that this absence of association in the high gamma band in ATN/DMTN could be due to a lack of power since broadband gamma is known to be spatially more focal than LFO⁴⁴. Our findings extend previous reports regarding the involvement of low frequency oscillations during reward-based tasks^{42,45}. The (negative) correlation of thalamic LFO with the outcome and the (positive) correlation with the expectation that were simultaneously observed after outcome display in ATN/DMTN are in accordance with the very notion of a prediction error signal. These results also mirror our previous finding that in the cortex, when we used broadband gamma activity as a neural index, we found a similar opponent association

between both components of prediction errors. Interestingly, the sign of the association reverted when comparing analyses based on broadband gamma and LFO in the cortex. This likely reflects the negative correlation existing between these two frequency bands, such as increased gamma power and decreased LFO accompanied local increase of the single-neuron firing rates^{46,47}.

Overall, whereas there is potentially high value and novelty in this dataset, there are significant concerns about the quality of the behavioral data, the specificity of the neural responses that would require significant reanalysis of the data. Similarly, the incomplete discussion would need to be significantly reworked.

We thank the Reviewer for many relevant and useful points that significantly helped us improve our manuscript.

Minor comments

Fig.1D and E – significance asterisks look like datapoints and are confusing – suggest differentiating in some other way (maybe coloring datapoints).

Data Points are now light gray dots for a better differentiation with asterisks in bold black.

How was the data baselined, if at all? Probably not critical for regression analysis but it should be specified; there is a vague mention of z-scoring but very little details. Was this done within f band, or some other way?

The data was baselined by z-scoring each point of the time frequency map by the power over a wide time epoch (10 seconds around each event of interest) within each frequency (as in Gueguen et al., 2021). We now specify this more clearly in the Methods section as follow:

Methods Line 353: Time-frequency power was converted into dB (decimal logarithm transformation) to improve the Gaussian distribution of the data and thereafter baselined using a trial wise z-score transform (using the average power in a 10 s time window centered on the event of interest), as previously described^{13,50}.

There is an asymmetry in learning from reward vs punishment (Fig.1) – do the authors have any interpretation or explanation for this?

There was no significantly detectable asymmetry in learning regarding the correct choice rate between reward and punishment conditions ($t(7) = 1.68, p = 0.14$), however reaction times were significantly longer in the punishment compared to the reward condition ($t(7) = 3.10, p = 0.017$). The absence of difference in accuracy, with a difference in reaction times has been repeatedly shown in this and similar tasks (see Fontanesi, CABN, 2019). One possibility that we discuss in the manuscript is that this increase in reaction time may reflect a manifestation of a Pavlovian bias according to which motor responses are inhibited by punishment expectations, irrespective of the appropriateness of the instrumental response. Alternatively, it could also reflect an extra-cognitive step characterizing punishment learning, as participants have to first identify the most punishing cue and then choose the alternative.

More details are needed regarding the number of electrodes in the sample. 8 patients received DBS leads with 4 contacts each, but there are 48 datasets in the final sample. Did some patients undergo bilateral implantation and had 8 contacts instead of 4? If all patients were bilateral, were any electrodes discarded? If so, what was the quality control criterion for exclusion?

All patients underwent a bilateral implantation of electrodes with 4 macro-contacts (Medtronic DBS electrode 3389). We rephrased the method and result section to clarify that preprocessing implied the use of a longitudinal bipolar montage so that there were 3 recording site (i.e., bipolar contact-pairs) by 2 hemispheres by 8 patients for a total of 48 recording sites. No electrodes were discarded, but artifacts and interictal discharges were discarded (see below). We clarified these points in the revised manuscript.

Methods Line 336: LFP signals were recorded with lead extensions connected to an EEG acquisition system (Micromed SD MRI, bandwidth 0.1–200 Hz, sampling rate 1024 or 2048 Hz). Each DBS electrode consisted of 4 contacts with a length of 1.5 mm, separated by 0.5 mm ((deep brain stimulation macro-electrode 3389, Medtronic, Minneapolis, US). Signal processing was performed using a longitudinal bipolar montage between the 3 adjacent pairs of contacts per electrode to maximize the sensitivity to local sources of LFP. Overall, 48 bipolar channels were recorded (3 contact-pairs/electrode × 2 hemispheres × 8 patients) using a commercial video-EEG monitoring system (System Plus, Micromed).

Methods Line 356: To remove artifacts and pathological interictal epileptiform discharge, we employed the following procedure for each recording site. A sliding window of 50 ms was employed within each event of interest (10 s time window centered on each event). Trials exhibiting a power that sporadically surpassed five times the standard deviation of the average signal were excluded. Consequently, an average of 6.1% of trials per patient were excluded from each epoching window. This exclusion rate primarily stems from two patients, who had an average of 20.6% of trials excluded.

Results Line 136: The neural activity of each recording site (n=48 bipolar channels, see Methods) was then regressed in the time-frequency domain against both expectation and outcome signals at different time points during the task.

Fig.2 a and b have no color scales, which makes it very hard to estimate the strength of the reported activations. And what do the contour lines represent? I assume increasing significance thresholds, but this is not explained.

Thanks, this is now fixed in the revised manuscript.

Author's first and last names order seems to change.

Thanks, this is now fixed.

1132 – “LFO oscillations” is redundant

We deleted the repetition.

REVIEWERS' COMMENTS

Reviewer #1 (Remarks to the Author):

Dear Authors,

Thank you for the detailed responses to my comments, which have all been addressed well. These findings relating to the role of the human thalamus in reward-processing, based on rarely available human intracranial electrophysiological recordings, will make an important and novel contribution to the literature.

Best regards,

Catherine Sweeney-Reed

Reviewer #2 (Remarks to the Author):

The authors have done a lot of work to adequately address the multiple issues found in the original submission. I will comment on a few of them:

- The author defend the behavioral performance of the patients as adequate. They add new analyses in which they show better accuracy towards the end of the behavioral task, which helps, but I found most useful in their response the reminder that these are surgical patients, with the difficulties that entails and the mention of comparable studies with similar performance rates. I think this makes the point appropriately – I found these references very useful and would suggest referencing them in the paper directly. I also appreciate that the authors now show that their QL model has lower AIC than a random choice model. A heuristic (win-stay/lose-switch) strategy may have been a better comparison, but this is an improvement in any case.
- To address concerns about the timecourse of LFO activations, the authors now include a new set of analysis from their earlier paper (Lopez-Persem 2020) in which they show very similar timecourses in cortical sites during the same task. I found this to be an excellent way to address this and I'm glad the authors leveraged their earlier dataset. Given these new data, the most likely explanation is a difference between LFO and gamma activity, as the authors acknowledge in their new Discussion line 240. This addition was brief and left me wanting a more in-depth discussion of this issue, but is understandable given space constraints. I'm satisfied on this point also.
- The expanded explanation on the time windows (Fig. R7) as well as the fact that the trials are interleaved greatly alleviates my concern about cross-trial bleed-through.
- The authors add new rationale for why low-level motor/visual activations could not account for their results, as well as a new exploratory analysis. I still think (and I suspect the authors would agree with me) that it is possible, or even likely, that thalamic activity reflects some of this low-level activations in addition to the learning signals. However, their rationale for why these are unlikely to explain their main results about the association between LFO and learning signals, even without a full exploration of the potential existence of motor/visual signals in thalamic activity, is solid. Perhaps a future direction to explore.
- Regarding the response to my question about the similarity in LFO activation between the reward and punishment conditions, the authors now state in the discussion: "Conversely, the association between thalamic LFO and outcomes (rewards and punishments) went in the same direction in both learning conditions. The similar directionality of outcome encoding may prima facie suggest that thalamic LFO signals behavioral saliency. However, we also note the positive association with reward expected value at the moment of choice and the moment of outcome (i.e., prediction error encoding) suggests that the expected reward value is encoded in thalamic LFO". I think what the authors are trying to say here is that there are differences in the learning signal generation, but they are encoded by the interplay between the expectation-related activity (which is different between reward/punishment) and the outcome-related activity (which is not). I think this is a great point to make, but the thought seems unfinished here – almost like a sentence explaining this outright at the end of this small paragraph is missing? I would suggest developing this further

or this important point may go unnoticed. Also, I suggest pointing to the relevant figures in this discussion text for clarity.

- I really appreciated the additional analysis about interindividual/interelectrode variation (new figure S6) which addresses my comments head on.
- The LFO power analyses are ok – I think a full-on time frequency representation would have been preferable here (and may speak to the existence of other activations not related to the learning signals), but this is a fairly minor point anyway.
- The new analyses across frequency bands, including gamma, are valuable and I was glad to see the lack of gamma association addressed in the discussion, even if briefly.
- My original point about the generation of RPEs (“...results implicate that there are separable bases for expected value and outcome value computations, but how these are combined to give rise to RPE signals (Fig.2) is ignored”) are very slightly alleviated by the new discussion on saliency etc. (see above), but mostly remains. This doesn’t invalidate the results, and maybe the authors plan to address more formally in the future, but I couldn’t help feeling that a more complete interpretation of this important point was within reach but not fully achieved. This remains a discussion point however.
- Minor point: last sentence of the abstract (“...reveal new insights into the neural computations that underlie this structure.”) is strangely phrased – suggest rewording e.g. to something like “underlie neural activation of this structure”.
- Also the title “The role of the thalamus in human reinforcement learning” is quite vague – I would strongly advice making it more specific to the results shown to better orient the reader. At least a mention to LFOs seems warranted.
- All my minor comments were appropriately addressed.

Overall, the authors have made a significant effort to address my earlier comments and as a result, even though a few smaller issues remain, these are mostly discussion issues more open to interpretation and the robustness of the data, findings and the main interpretation is significantly improved.

Reviewer #1

Dear Authors,

Thank you for the detailed responses to my comments, which have all been addressed well. These findings relating to the role of the human thalamus in reward-processing, based on rarely available human intracranial electrophysiological recordings, will make an important and novel contribution to the literature.

Best regards, Catherine Sweeney-Reed

We thank Dr. Catherine Sweeney-Reed for her supportive comments: that's great to hear!

Reviewer #2

The authors have done a lot of work to adequately address the multiple issues found in the original submission. I will comment on a few of them:

We thank the Reviewer 2 for this supportive comment and for the effort to underlie how we responded to previous issues in the following sections.

R2.1. The author defend the behavioral performance of the patients as adequate. They add new analyses in which they show better accuracy towards the end of the behavioral task, which helps, but I found most useful in their response the reminder that these are surgical patients, with the difficulties that entails and the mention of comparable studies with similar performance rates. I think this makes the point appropriately – I found these references very useful and would suggest referencing them in the paper directly. I also appreciate that the authors now show that their QL model has lower AIC than a random choice model. A heuristic (win-stay/lose-switch) strategy may have been a better comparison, but this is an improvement in any case.

We added the references to comparable behavioral results directly in the manuscript:

Results Line 133-136: The observed rate of correct choices in those pharmacoresistant epileptic patients were comparable to similar studies in the field, such as those conducted with similar tasks in other clinical cohorts, such as Parkinson (~60%)³², Tourette (~63%)³³ and, more recently, epileptic patients (~70%)¹³.

R2.2. To address concerns about the timecourse of LFO activations, the authors now include a new set of analysis from their earlier paper (Lopez-Persem 2020) in which they show very similar timecourses in cortical sites during the same task. I found this to be an excellent way to address this and I'm glad the authors leveraged their earlier dataset. Given these new data, the most likely explanation is a difference between LFO and gamma activity, as the authors acknowledge in their new Discussion line 240. This addition was brief and left me wanting a more in-depth discussion of this issue, but is understandable given space constraints. I'm satisfied on this point also.

We thank the reviewer 2 for pushing us to include a new set of analysis from our earlier cortical iEEG data-set.

R2.3. The expanded explanation on the time windows (Fig. R7) as well as the fact that the trials are interleaved greatly alleviates my concern about cross-trial bleed-through.

We're delighted that this alleviates the reviewer's concerns about this previous issue (that is now fixed).

R2.3. The authors add new rationale for why low-level motor/visual activations could not account for their results, as well as a new exploratory analysis. I still think (and I suspect the authors would agree with me) that it is possible, or even likely, that thalamic activity reflects some of this low-level activations in addition to the learning signals. However, their rationale for why these are unlikely to explain their main results about the association between LFO and learning signals, even without a full exploration of the potential existence of motor/visual signals in thalamic activity, is solid. Perhaps a future direction to explore.

We agree that these remarks are good food for thoughts and possibly a future direction to explore.

R2.4. Regarding the response to my question about the similarity in LFO activation between the reward and punishment conditions, the authors now state in the discussion: “Conversely, the association between thalamic LFO and outcomes (rewards and punishments) went in the same direction in both learning

conditions. The similar directionality of outcome encoding may prima facie suggest that thalamic LFO signals behavioral saliency. However, we also note the positive association with reward expected value at the moment of choice and the moment of outcome (i.e., prediction error encoding) suggests that the expected reward value is encoded in thalamic LFO". I think what the authors are trying to say here is that there are differences in the learning signal generation, but they are encoded by the interplay between the expectation-related activity (which is different between reward/punishment) and the outcome-related activity (which is not). I think this is a great point to make, but the thought seems unfinished here – almost like a sentence explaining this outright at the end of this small paragraph is missing? I would suggest developing this further or this important point may go unnoticed. Also, I suggest pointing to the relevant figures in this discussion text for clarity.

Thank you for your careful review. We now state more clearly the idea by adding a final sentence explaining this outright at the end of this paragraph while we also point to the relevant figures (Suppl. Figure S5) in this section to improve clarity.

Discussion Line 250: Conversely, the association between thalamic LFO and outcomes (rewards and punishments) went in the same direction in both learning conditions (Fig. S5). The similar directionality of outcome encoding may prima facie suggest that thalamic LFO signals behavioral saliency. Yet, the (positive) correlation between thalamic LFO with the reward outcome and the (negative) correlation with the reward expectation were both observed after outcome display (Fig. 3). These opponent associations are in accordance with the very notion of reward prediction error, as it demonstrates a straightforward neural implementation of the difference between the outcome and the expectation components of the teaching signal. Furthermore, the stronger association between punishment expectation compared to reward expectation at the time of choice (Fig. 2) also speaks against the idea that saliency alone could explain the current results.

R2.5. I really appreciated the additional analysis about interindividual/interelectrode variation (new figure S6) which addresses my comments head on.

That's nice to read 😊.

R2.6. The LFO power analyses are ok – I think a full-on time frequency representation would have been preferable here (and may speak to the existence of other activations not related to the learning signals), but this is a fairly minor point anyway.

Thank you for agreeing for LFO power analyses as sufficient for the aim of this study.

R2.7. The new analyses across frequency bands, including gamma, are valuable and I was glad to see the lack of gamma association addressed in the discussion, even if briefly.

We believe this additional information is valuable for the field.

R2.8. My original point about the generation of RPEs (“...results implicate that there are separable bases for expected value and outcome value computations, but how these are combined to give rise to RPE signals (Fig.2) is ignored”) are very slightly alleviated by the new discussion on saliency etc. (see above), but mostly remains. This doesn't invalidate the results, and maybe the authors plan to address more formally in the future, but I couldn't help feeling that a more complete interpretation of this important point was within reach but not fully achieved. This remains a discussion point however.

We agree with the Reviewer and we slightly edited the discussion section to more directly discuss how thalamic LFO relate to the notion of reward prediction errors.

Discussion Line 250: Conversely, the association between thalamic LFO and outcomes (rewards and punishments) went in the same direction in both learning conditions (Fig. S5). The similar directionality of outcome encoding may prima facie suggest that thalamic LFO signals behavioral saliency. Yet, the (positive) correlation between thalamic LFO with the reward outcome and the (negative) correlation with the reward expectation were both observed after outcome display (Fig. 3). These opponent associations are in accordance with the very notion of reward prediction error, as it demonstrates a straightforward neural implementation of the difference between the outcome and the expectation components of the teaching signal. Furthermore, the stronger association between punishment expectation compared to reward expectation at the time of choice (Fig. 2) also speaks against the idea that saliency alone could explain the current results.

R2.9. Minor point: last sentence of the abstract (“...reveal new insights into the neural computations that underlie this structure.”) is strangely phrased – suggest rewording e.g. to something like “underlie neural activation of this structure”.

The editorial team proposed a revised abstract including a modification of this last sentence.

Abstract Line 32: Reinforcement-based adaptive decision-making is believed to recruit fronto-striatal circuits. A critical node of the fronto-striatal circuit is the thalamus. However, direct evidence of its involvement in human reinforcement learning is lacking. We address this gap by analyzing intra-thalamic electrophysiological recordings from eight participants while they performed a reinforcement learning task. We found that in both the anterior thalamus (ATN) and dorsomedial thalamus (DMTN), low frequency oscillations (LFO, 4-12 Hz) correlated positively with expected value estimated from computational modeling during reward-based learning (after outcome delivery) or punishment-based learning (during the choice process). Furthermore, LFO recorded from ATN/DMTN were also negatively correlated with outcomes so that both components of reward prediction errors were signaled in the human thalamus. The observed differences in the prediction signals between rewarding and punishing conditions shed light on the neural mechanisms underlying action inhibition in punishment avoidance learning. Our results provide insight into the role of thalamus in reinforcement-based decision-making in humans.

R2.10. Also, the title “The role of the thalamus in human reinforcement learning” is quite vague – I would strongly advise making it more specific to the results shown to better orient the reader. At least a mention to LFOs seems warranted.

We changed the title to make it more specific, as also suggested by the editorial team

Title Line 1: Human thalamic low-frequency oscillations correlate with expected value and outcomes during reinforcement learning

R2.11. All my minor comments were appropriately addressed.

Thank you for those comments.

Overall, the authors have made a significant effort to address my earlier comments and as a result, even though a few smaller issues remain, these are mostly discussion issues more open to interpretation and the robustness of the data, findings and the main interpretation is significantly improved.

Again, we thank the Reviewer 2 for the supportive comments which allowed a clear improvement of the quality of the manuscript.